# Flexibility of intrinsic neural timescales during distinct behavioral states
Yasir Çatal [1,2] ✉, Kaan Keskin[2,3,4,8], Angelika Wolman [1,2,8], Philipp Klar[5,6,8], David Smith[2,7] & Georg Northoff [1,2]

Recent neuroimaging studies demonstrate a heterogeneity of timescales prevalent in the brain's ongoing spontaneous activity, labeled intrinsic neural timescales (INT). At the same time, neural timescales also reflect stimulus- or task-related activity. The relationship of the INT during the brain's spontaneous activity with their involvement in task states including behavior remains unclear. To address this question, we combined calcium imaging data of spontaneously behaving mice and human electroencephalography (EEG) during rest and task states with computational modeling. We obtained four primary findings: (i) the distinct behavioral states can be accurately predicted from INT, (ii) INT become longer during behavioral states compared to rest, (iii) INT change from rest to task is correlated negatively with the variability of INT during rest, (iv) neural mass modeling shows a key role of recurrent connections in mediating the rest-task change of INT. Extending current findings, our results show the dynamic nature of the brain's INT in reflecting continuous behavior through their flexible rest-task modulation possibly mediated by recurrent connections.

In order to survive in an ever-changing environment with a wide range of complexity on multiple spatial and temporal scales, the brain must coordinate the behavior of the organism in a flexible way. The stimuli from the external world comprise a wide variety of timescales reflecting distinct durations: for example, each dialog is an interplay of syllables, words, and sentences in an increasing timescale, carrying meaning and demanding a reaction[1–5]. To process such temporally complex information, the brain utilizes a heterogeneity of timescales in its own neural activity[1,3–6]. The concept of timescale in this context is defined as the duration of the temporal window within which a particular stimulus is processed with others in the same time window—this has been subsumed under the umbrella term "intrinsic neural timescales" (INT)[1,2,7,8].

Recent neuroimaging work has shown that INT exhibit changes during various aspects of cognitive processing. Examples of tasks showing INT changes include reward[8], self[9], narrative construction[10], consciousness[11,12] and music listening[13]. In contrast, studies on the changes of INT during continuous behavior are scarce. Current work indicates that INT become longer with increasing levels of task engagement and attention during behavior[14–18]. At the same time, the INT are already present in the brain's spontaneous activity where no specific tasks, stimuli or behavior are initiated[4,19–21]. How the INT in continuous behavioral states relate to the

timescales of the brain's spontaneous activity remains yet unclear[22,23]. Addressing this question is the goal of our study.

First and foremost, we may want to do some conceptual homework. What do we mean by the notions of behavioral state and brain state? A behavioral state refers to a distinct pattern of locomotor activity, cognition, and physiological activity that an organism exhibits in response to internal and external stimuli within a particular environmental context[24]. While an agreed upon definition of a brain state is currently lacking[25], nonetheless, we assume the following more operational definition: a brain state is characterized by a widely distributed pattern of activity[26] which can be specified by its spatial pattern (like its topographic distribution) and temporal pattern (like its changes or non-changes over time along different timescales). We here focus specifically on the temporal pattern of brain states during continuous naturalistic behavior which by itself, as implied by its definition, may also be featured by a particular temporal pattern with the involvement of various timescales.

Brain states during different tasks and behavioral states are set relative to the brain's spontaneous activity and its particular temporal pattern—this raises the question for their relationship. Such focus on the modulation of task INT by rest INT is motivated by a wide range of studies demonstrating that the dynamics of rest or pre-stimulus state can modulate the brain's

[1]Mind, Brain Imaging and Neuroethics Research Unit, University of Ottawa, Ontario, ON, Canada. [2]University of Ottawa Institute of Mental Health Research, Ottawa, ON, Canada. [3]Department of Psychiatry, Ege University, Izmir, Turkey. [4]SoCAT Lab, Ege University, Izmir, Turkey. [5]Faculty of Mathematics and Natural Sciences, Institute of Experimental Psychology, Heinrich Heine University of Düsseldorf, Düsseldorf, Germany. [6]Institute of Neuroscience and Medicine, Brain & Behaviour (INM-7), Research Centre Jülich, Jülich, Germany. [7]Department of Cellular and Molecular Medicine, University of Ottawa, Ottawa, ON, Canada. [8]These authors contributed equally: Kaan Keskin, Angelika Wolman, Philipp Klar. ✉e-mail: catalyasir@gmail.com

response to an external stimulus. Various studies in functional magnetic resonance imaging (fMRI)[27–40], electroencephalography (EEG)[9,41–48] and single-unit neural recordings[49–51] show an active modulation of the task-related activity's amplitude, variability, and functional connectivity by the brain's spontaneous activity (see also[52,53] for a review). Although it should be mentioned that other findings noted no relationship or the divergence between spontaneous and task-evoked activity[54–58]: these studies show that the dimensionality of the resting state dynamics differs greatly from task evoked states with little or no correlation of rest and task states—in that case, task-related and spontaneous activity are orthogonal to each other. Whether this difference is physiological, as related to distinct kinds of tasks, or methodological, as related to different analyses methods[41,42] remains yet to be clarified and tested. We here follow the analyses methods of the first stance, the relationship of spontaneous and task activity and develop our hypotheses accordingly. Following this line of argument, how the changes in INT during behavioral or task-induced states relate to the spontaneous activity has, to our knowledge, not yet been studied. Building on these observations of rest-task modulation of neural activity in its amplitude, variablity and functional connectivity, we ask whether the INT modulate the transition from rest to continuous behavioral task states.

Our paper has four specific aims, raising distinct questions. First, we question whether the INT are indeed behaviorally relevant. We use two behavioral paradigms: calcium imaging data from spontaneously behaving mice[59] and human EEG data where participants track their internal states using a mouse cursor while listening to autobiographical (self-related) and non-autobiographical (non-self/other related) narratives[60]. Based on the behavioral recordings of mice, we demarcate time windows of activity, such as sustained rest and locomotion, denoting the different behavioral states over the course of time. While in the human EEG, we divide the recordings into distinct time windows during rest and task. In both datasets, we use these temporal windows to estimate their INT; the latter are estimated by using the decay rate of the autocorrelation function which measures the duration of temporal autocorrelation in a time-series[8,19,21,60–63] in neuroimaging data[8,62]. We raise the question whether the behavioral state of the organism can be predicted from the INT across the whole brain's neural activity. For this purpose, we train SVMs to classify the behavioral states in mice (onset of locomotion, locomotion, offset of locomotion, initial rest, sustained rest) and humans (rest, autobiographical narrative, non-autobiographical narrative) according to their whole brain INT estimates. Following the INT differences in rest and event-related discontinuous task states, we hypothesize that the INT can also distinguish distinct more continuous, and naturalistic behavioral states in both mice (rest and its distinct forms, locomotion) and human data (rest, self, and non-self).

Our second question is whether there is a flexible change in the INT from rest to behavioral states. Given the previous findings of an increasing duration of INT during behavioral states[17] and increased attention[15,16], we hypothesize longer INT during behavioral states compared to the resting state. To address this question, we compare the INT in rest and locomotion states in mice as well as during rest and task states in humans.

The third question is related to the relationship of rest INT with the INT changes during behavior: can the resting state variability account for their rest-behavior change, and if so, in what way? Previous studies mentioned above have established that neuronal activity at task is modulated by the resting state activity with a special role for the variability of the amplitude[36,41,42,44]. Extending beyond the amplitude and its variability, we aim to extend these observations for the case of timescales by correlating the variability of INT during the resting state with the percent change of INT from rest to behavioral states.

Our fourth and last question investigates the potential dynamic mechanisms required for the assumed rest-behavior modulation of INT in simplified mathematical models. Previous theoretical work[16,64] on the relationship between local connectivity profile and INT indicate that in the critical regime of the neuronal population, small changes in recurrent connections enable flexibility in the INT. Based on this insight, we aimed to replicate our empirical results in a large-scale biophysical model in order to investigate the role of recurrent connections on the flexibility of INT changes during the transition from rest to task states. We used firing rate model with a connectivity matrix derived from diffusion tensor imaging (DTI) of healthy humans[65]. We operationalized the behavioral state by stimulating the regions which increases their firing rate and calculated the INT in non-overlapping time windows for every brain region in both rest and stimulated states.

On the more methodological side, we should note that while the psychological requirements of the two tasks (motor outputs versus self-evaluation) differ from each other, they nevertheless share a fundamental factor that is essential to the INT: both tasks require continuous, that is, non-interrupted behavior along different durations, e.g., timescales; this distinguishes the task states in both mice and humans from their respective resting state that does not require such continuous non-interrupted activity—this is measured by quantifying the temporal continuity of their underlying neural activity by the INT.

Our findings reveal that the continuous behavioral states can be predicted with high accuracy from the INT across the whole brain in both mice and human data. Replicating and extending previous findings, we found longer INT during both behavioral states, e.g., locomotion in mice and tracking internal states in humans. We observed that the changes in INT from rest to behavioral states in both mice and humans are negatively correlated with the variability of the INT during rest. Our computational model replicated the empirical data in both the increased INT in the stimulated state (compared to the non-stimulated state) and the negative correlation between resting state variability and INT change. We observed that by increasing the recurrent connections in the model, rest-stimulated state changes in INT also increased up to a point of saturation. In contrast, the decrease in recurrent connections diminished the degree of INT rest-task change. While in the case of absent recurrent connections, the rest task change of INT was negligible. Together, our findings show the flexible nature of the INT during continuous naturalistic behavioral states through their rest-task interaction possibly mediated by recurrent connections in the brain.

## Results
### Behavioral states in mice are distinguished by their intrinsic neural timescales

We started our analysis with segmenting the neural activity into different periods of behavioral states. The data was segmented according to the following heuristic[59]: onset of locomotion (onset): locomotion, offset of locomotion (offset), rest after the end of locomotion (initial rest), rest 40 s after the end of locomotion (sustained rest) (Fig. 1a). These distinct behavioral segments will be called time windows for the rest of the paper. We estimated INT via fitting an exponential decay function with a decay rate $\frac{1}{\tau}$ and extracting the parameter $\tau$ which corresponds to how long does it take for the autocorrelation function to reach $\frac{1}{e}$. Fig. 2a shows the $\tau$ values across the brain for one of the mice. The same figures for the other mice can be found in Supplementary Fig. 1.

To test whether the behavioral states can be distinguished by their underlying timescales, we used support vector machine (SVM) with a radial basis kernel. We argue that if various behavioral states can be classified on the basis of their $\tau$ values by SVM, it would show that the INT can distinguish different behavioral states. We performed the SVM by concatenating all the $\tau$ values across time windows (822 time windows, 92 ROIs as features), runs, and mice; giving us 92 features (one for each region of interest (ROI)). The models were trained with nested cross validation (10 inner, 10 outer folds). The confusion matrix for test data aggregated across folds can be seen in Fig. 2b. In all behavioral states, SVM performed higher than random chance (20% for 5 states), between 81% for locomotion and 94% for sustained rest; and showed an overall accuracy of 86%. The one-versus-all ROC curve averaged across folds is given in Fig. 2c, showing that for every state, SVM reached an AUC of more than 0.9, indicating that false positives and false negatives are relatively low. The results for individual folds can be found in Supplementary Figs. 2 and 3. These results demonstrate that INT

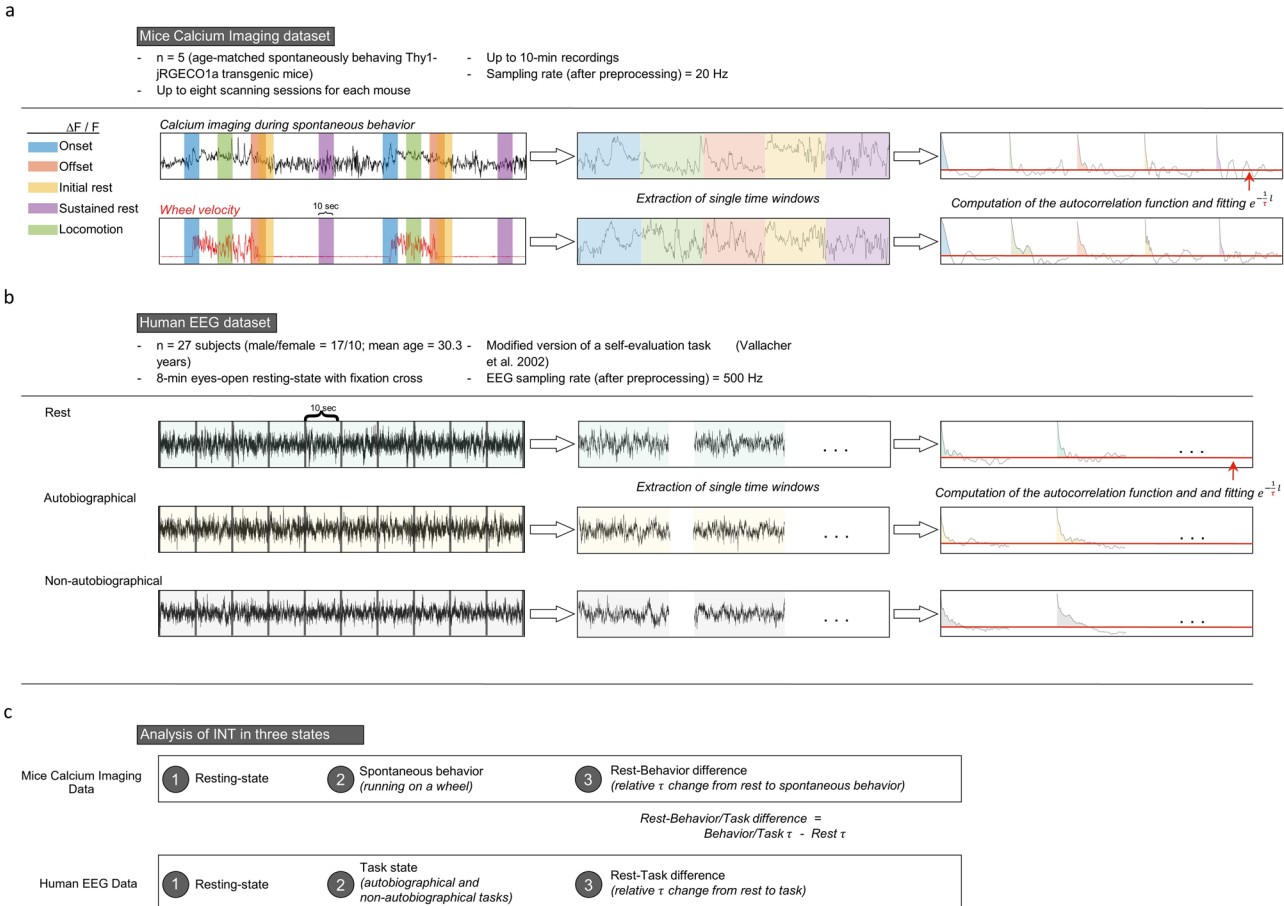

**Fig. 1 | Outline of the data and analysis. a** We used calcium imaging recordings of spontaneously behaving mice. Based on behavioral recording of wheel velocity, we segmented the data for various behavioral states. Each segment is 10 s long. We calculated the autocorrelation function (ACF) and estimated intrinsic neural timescales (INT) by fitting an exponential decay function of the form $e^{-\frac{1}{\tau}l}$ and extracted the parameter $\tau$ (see methods for a detailed explanation). **b** Healthy humans were recorded with EEG in three conditions: resting state, listening to an autobiographical narrative recorded by themselves, and listening to a non-autobiographical narrative recorded by somebody else. In listening conditions, participants tracked their internal state using the mouse cursor. Continuous recordings were segmented into 10 s windows. In each window, we calculated the $\tau$ in the same way as in mice. **c** We analyzed the $\tau$ values in rest, task/behavior, and investigated the rest-task or behavior changes.

allow distinguishing between different behavioral states. Same analysis using ACW-0 (the lag where autocorrelation function reaches 0) instead of $\tau$ can be seen in Supplementary Fig. 4 for averages across folds and Supplementary Figs. 5 and 6 for individual folds. ACW-0 results again show high accuracy and AUC values for ROC curves.

In addition to SVM, we also performed logistic regression as a prediction task as opposed to classification of SVM. We fitted the logistic regression model in the same way as in SVM: using 10-fold nested cross validation. Supplementary Figs. 7, 8, 9 show the average ROC curve and aggregate confusion matrix, that is, confusion matrices for individual folds and ROC curves for individual folds respectively. We evaluated the model's performance on test data as in SVM. The regression model showed an overall accuracy of 86%. For every state, the ROC curves have an AUC above 0.9.

Together, we show that the INT, measured as the decay rate of the autocorrelation function, can distinguish different behavioral states in the mice. This suggest a close relationship between INT and behavior which shall be explored in more detail in the following.

**Changes in intrinsic neural timescales from rest to behavior**

For the rest of the paper, we will focus on sustained rest and locomotion states. We started our investigation by comparing the means of $\tau$s during these two states. We divided our analysis into two parts: comparing the mean across time windows for each ROI and comparing the mean across

ROIs for each time window. For the mean across time windows, we averaged $\tau$ values across windows for each ROI. For mean across ROIs, we averaged the $\tau$ values across the brain for each window of sustained rest and locomotion.

Figure 3 shows the comparison of $\tau$ values averaged across windows for each ROI. The difference between mean $\tau$ values for sustained rest and locomotion in all mice is shown in Fig. 3a (Wilcoxon test ($n = 460$ for each group, $z = 25.07$, $p < 0.001$, $r = 0.95$)). The same comparisons for each of the individual mice is shown in Fig. 3b ($n = 92$). In each case, we see an increase in the mean of $\tau$ for each ROI from sustained rest to locomotion. Furthermore, we performed a mixed effects analysis treating mice labels and ROIs as random effects and the state of the mouse (locomotion versus sustained rest) as fixed effect and $\tau$ values as independent variable. We have found a significant effect for state ($F_{1,823} = 2349.376$, $p < 0.001$). Same analyses for ACW-0 instead of $\tau$ are presented in Supplementary Fig. 10, showing higher ACW-0 in task state compared to rest in each mice.

In addition to INT, we also compared mean and standard deviation (SD) of the neural activity to characterize the neural activity more thoroughly. One mean and SD value was extracted from each ROI per 10 s time window of activity, in the same way as the INTs were calculated. We averaged the neural activity across these windows and compared them between sustained rest and locomotion conditions. The results are presented in Supplementary Fig. 11. In short, we observed higher mean of activity in locomotion compared to sustained rest ($z = 25.64$, $p < 0.001$, $r = 0.98$) and

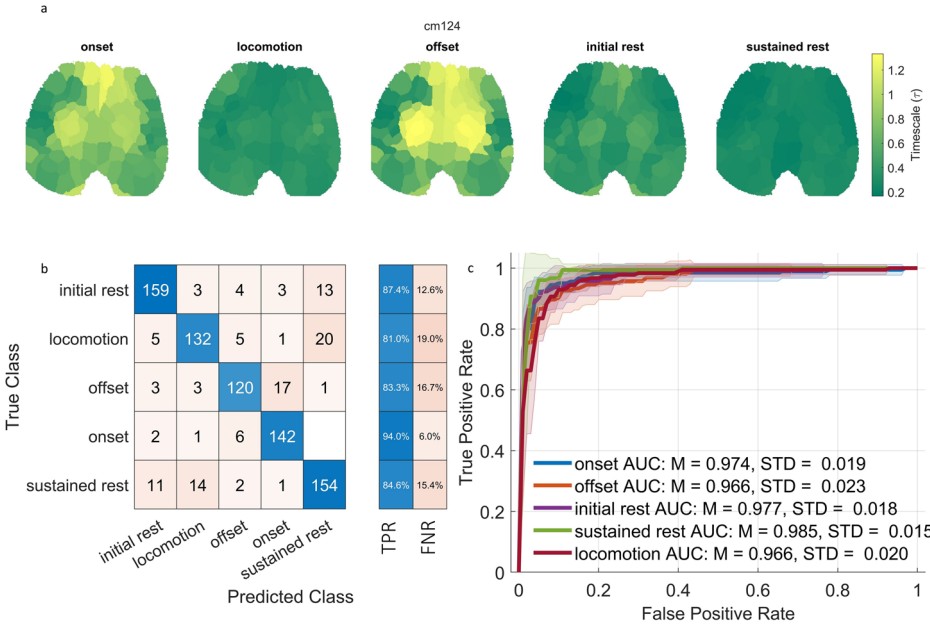

**Fig. 2 | Classification learning for behavioral states.** **a** The averaged $\tau$ values across the brain for one of the mice. The same figure for other mice can be found in Supplementary Fig. 1. **b** We trained support vector machines to classify the behavioral states of mice based on the $\tau$ values across the brain with nested cross validation for hyperparameter tuning (10 inner, 10 outer folds). **b** Shows the confusion matrix for test data, aggregated across folds. **c** Receiver operating characteristic curve for the test data using the trained support vector machine. The data ROC curves from 10 folds were averaged and the standard deviation across the folds were indicated as shading. Abbreviations: TPR True positive rate, FNR False negative rate, AUC Area under the curve, M Mean, STD Standard deviation.

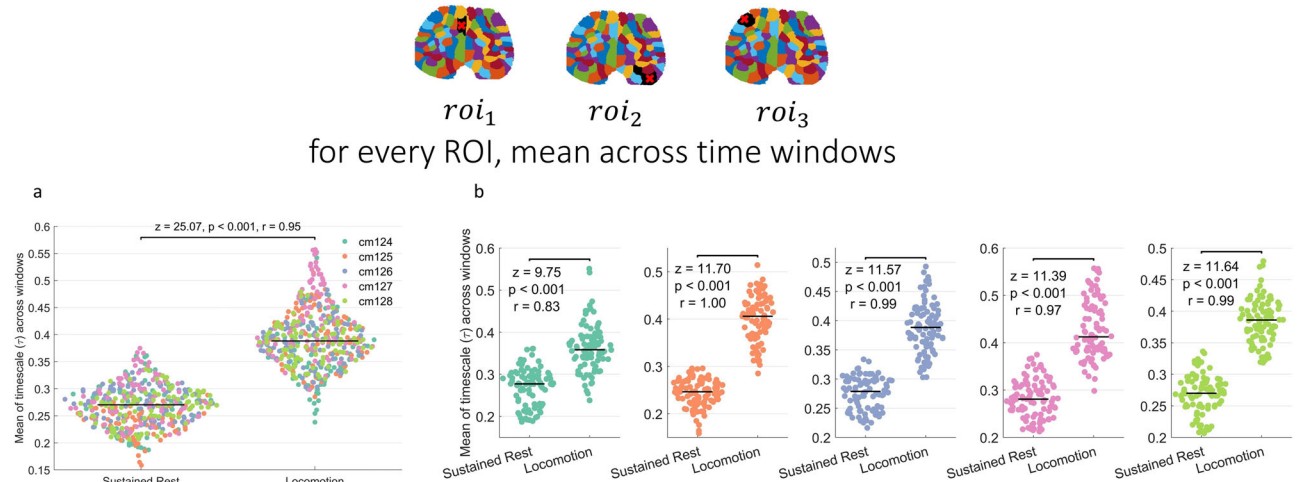

**Fig. 3 | Comparison of mean $\tau$s averaged across time for each region of interest (ROI). a** Comparison of $\tau$ values between sustained rest and locomotion states in all mice. **b** Comparison for each mouse. Each dot in the figure represents the $\tau$ value of one ROI. Colors denote individual mice.

higher SD in sustained rest compared to locomotion ($z = 15.53$, $p < 0.001$, $r = -0.59$).

It can be argued that the relationship between INT and behavioral states is a spurious one, possibly mediated by some other aspect of neuronal dynamics. Indeed, previous research has shown that neural variability and oscillations also reflect the behavioral states[46,66–74]. To demonstrate that the relationship between INT and behavioral states is not spurious, we present a two-step analysis: (1) showing potential correlations between INT and variance (for mice data); INT and power bands (for human EEG data); (2) showing that the relationship between INT and behavioral state does not change significantly when conditioned on the standard deviation (SD, normalized variance) of the data[75]. To achieve the second step, we first perform a logistic regression with INT as predictor and behavioral state as outcome, then do a second regression with ACW and SD and compare the regression coefficients using the statistical test developed by Clogg et al.[76]. The results for the correlation are presented in Supplementary Fig. 12. We observe a negative correlation between SD and INT (Spearman's $\rho = -0.107$, $p = 0.047$). The complete results for the logistic regressions are presented in Supplementary Tables 1 and 2. Clogg test do not show a

significant change in the regression coefficient ($z = 0.0162$, $p = 0.987$). While we cannot say that the regression coefficient stays the same due to the logic of null hypothesis significance testing, we can rule out the claim that the predictive value of ACW is a spurious one in that it solely depends on the measures of the neuronal dynamics like variability we tested, that is, that the relationship of INT and behavioral state is spurious.

As a next step, we classified the regions of interest in the following groups: frontal; somatosensory 1, 2, 3; visual and motor for left and right hemispheres, giving us 12 groups of ROI in total similar to (Shahsavarani et al.[59]). This classification is visualized in Supplementary Fig. 13. We compared rest and task $\tau$ values according to this classification. The results are shown in Supplementary Fig. 14. We observe a significant increase in $\tau$ in all ROI groups.

Before moving on, we also compared the early and late periods of sustained rest and locomotion in a recording. We picked recordings that have more than one sustained rest period and compared the first sustained rest and the last one. We did the same analysis for locomotion as well. In both analyses, no significant differences between early and late periods were found (Supplementary Fig. 15).

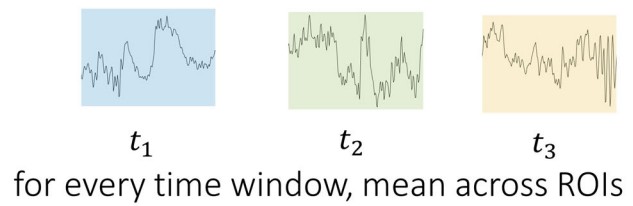

for every time window, mean across ROIs

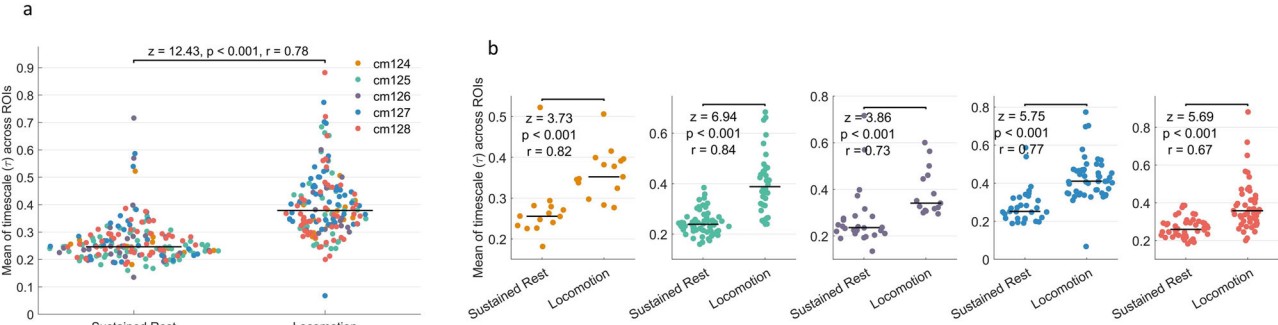

**Fig. 4 | Comparison of mean $\tau$s averaged across regions of interest (ROIs) for each time window. a** Comparison of $\tau$ values between sustained rest and locomotion states in all mice. **b** Comparison for each mouse. Each dot in the figure represents $\tau$ value of one time window. Colors denote individual mice.

We then moved on to compare the INT rest task changes in the whole brain. For each window, we averaged the $\tau$ values across ROIs. Fig. 4a shows the change of mean across the brain for each window, showing a significant increase in brain-wide $\tau$ ($n = 182$ for sustained rest, $n = 163$ for locomotion; $z = 12.43$, $p < 0.001$, $r = 0.78$). The comparisons for each of the individual mice are shown in Fig. 4b showing that this increase holds in each mouse (cm124: $n = 14$ for sustained rest, $n = 15$ for $z = 3.73$, $p < 0.001$, $r = 0.82$; cm125: $n = 63$ for sustained rest, $n = 36$ for locomotion, $z = 6.94$, $p < 0.001$, $r = 0.84$; cm126: $n = 26$ for sustained rest, $n = 15$ for locomotion, $z = 3.86$, $p < 0.001$, $r = 0.73$; cm127: $n = 31$ for sustained rest, $n = 48$ for locomotion, $z = 5.75$, $p < 0.001$, $r = 0.77$; cm128: $n = 48$ for sustained rest, $n = 49$ for sustained rest, $z = 5.69$, $p < 0.001$, $r = 0.67$). A mixed effects model treating the state of the mice (locomotion versus sustained rest) as fixed effect and mouse IDs as random effect found a significant effect for the states ($F_{1, 341.79} = 164.225$, $p < 0.001$). The analysis for ACW-0 can be found in Supplementary Fig. 16, showing higher ACW-0 in locomotion again.

Additionally, we compared the pupil size between sustained rest and locomotion states to assess the behavioral changes that go along with changes in $\tau$. For each time window, we averaged the pupil size across time. Supplementary Fig. 17 shows that the pupil size is significantly larger during locomotion compared to the sustained rest ($n = 182$ for sustained rest, $n = 163$ for locomotion; $z = 14.6$, $p < 0.001$, $r = 1.19$). Together, we show increased $\tau$ from sustained rest to locomotion state for both individual ROIs and across the whole brain. This increase in $\tau$ goes together with an increased pupil size. (Supplementary Fig. 17)

**Resting state variability of intrinsic neural timescales shapes their rest-locomotion change**

Motivated by the studies showing that the neural activity during the behavior is influenced by the variability of prestimulus/rest activity, we sought to analyze the effect of INTs of rest on INTs during locomotion. In order to assess the effect of rest on rest-locomotion change, we correlated the rest variability of $\tau$ values across time windows with the percent change of $\tau$ values from rest to locomotion. We calculated the rest-locomotion percent change of $\tau$ as $(\frac{locomotion\ value - rest\ value}{rest\ value}) x100$. In Fig. 5a, we correlated the resting state variability with the percent change of $\tau$, seeing a negative correlation (Spearman's $\rho(458) = -0.34$, $p < 0.001$).

Figure 5b shows the distribution of INT rest variability and the rest-locomotion INT percent change across the whole brain. Fig. 5c shows this relationship for each of the mice ($n = 92$ in each plot), showing that all mice exhibit such negative correlation. Together, we demonstrate that the $\tau$ resting

state variability exert a negative modulation on the behavior related changes of $\tau$ i.e., negative rest task modulation of $\tau$. Higher variability in resting state $\tau$ indexing its dynamic repertoire leads to lower changes in $\tau$ from rest to locomotion. Same analyses for ACW-0 instead of $\tau$ are presented in Supplementary Fig. 18, showing negative correlation for each of the mice.

**Replication of the mice results in a human EEG dataset of self-evaluation**

We also analyzed a human EEG dataset where participants performed a modified self-evaluation task[77]. In this task, healthy humans listen to an autobiographical narrative (self-related) recorded by themselves or a personally unrelated narrative (non-self/other) recorded by the researchers. During the listening, the participants actively track their internal states by moving the cursor on the screen (Fig. 1b, see "methods" and ref. 60 for details). Both mice and humans share self-initiated behavior with either locomotion or moving the cursor on the screen. However, they differ in the level of spontaneity since the mice's behavior was purely spontaneous without any external cue while the humans were instructed to use the cursor according to how they felt during listening to narratives. Like mice data, we calculated one $\tau$ value from every 10-s non-overlapping window.

Figure 6a shows the distribution of $\tau$ values across the scalp averaged across time windows and 21 subjects (2961 time windows and 64 channels as features). We started our analysis by replicating the SVM findings with a radial basis kernel. The SVMs were trained using nested cross validation with 10 inner and 10 outer folds. The confusion matrix for test data aggregated across folds is given in Fig. 6b, showing that the SVM is reaching an accuracy of at least 72% while random chance is 33.3%. Overall accuracy is 75%. ROC curves averaged across folds are given in Fig. 6c, showing a minimum AUC of 0.85, indicating that the different states (rest, self, non-self) can indeed be distinguished by their $\tau$ values. Confusion matrices and ROC curves for individual folds can be found in Supplementary Figs. 19 and 20. Using ACW-0, we replicated our results of $\tau$ (Supplementary Fig. 21 for confusion matrix aggregated across folds and ROC curve averaged across folds; Supplementary Figs. 22 and 23 for confusion matrices and ROC curves respectively for individual folds).

We additionally used logistic regression in addition to SVM. Supplementary Figs. 24, 25, 26 show the results for average ROC curve and aggregate confusion matrix, confusion matrices for individual folds and ROC curves for individual folds respectively. The overall accuracy of the model evaluated in the test set is 58%. The ROC curves have an AUC value above 0.7 for all states.

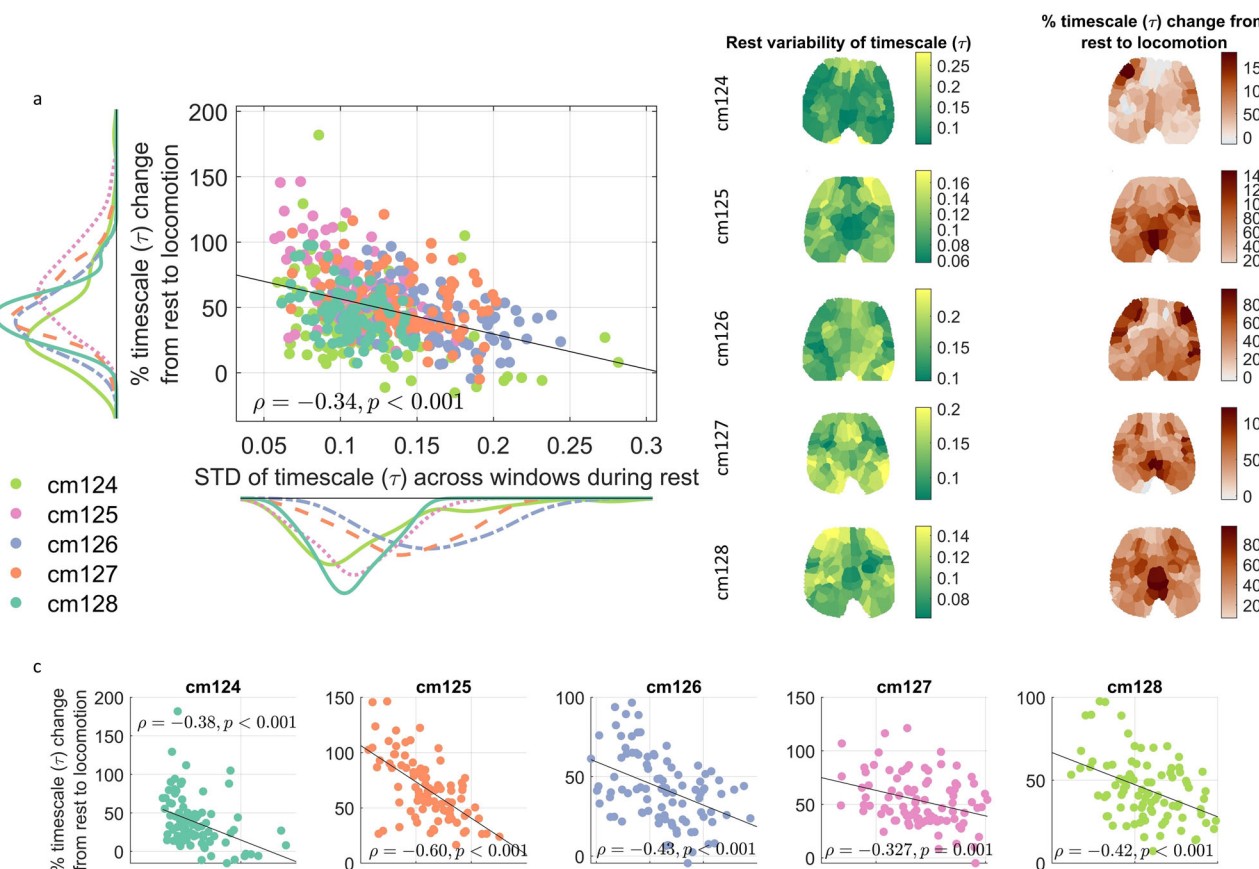

**Fig. 5 | Rest-Behavior modulation of INT. a** We calculated the variability of $\tau$s across time windows. To calculate the change of $\tau$s from rest to locomotion, we averaged the percent change of $\tau$s in each of the aforementioned windows. We correlated the variability of $\tau$s during sustained rest with the $\tau$ percent change from rest to locomotion. Each dot denotes one region of interest in one recording session. **b** The distribution of the variability of $\tau$ during rest and the percent change of $\tau$ from rest to locomotion across the whole brain. **c** Same as (**a**) for each of the individual mice.

As in the mice, the findings suggest that INT, as measured by $\tau$, allow distinguishing distinct behavioral states in human EEG.

We proceeded with the rest-task comparison of INT in the human EEG data. Fig. 7a shows that during the self-stimulus, INT are the longest, followed by other and rest ($n = 1344$ for each group). A mixed effects model treating the state (rest/self/other) as fixed effect, channels and subjects as random effect a significant effect for the state ($F_{2, 3946} = 134.199$, $p < 0.001$). This pattern is also preserved in the mean across channels for every time window ($n = 987$ for each group) (Fig. 7b: mixed effects with state as fixed effect and subjects as random effect: $F_{2,2938} = 99.193$, $p < 0.001$). Multiple comparisons using Wilcoxon tests and effect size calculations can be seen in Fig. 7 panels a and b. These results indicate that the states with self-initiated behavior, e.g., moving the cursor while listening to the narrative are characterized by longer INT than the resting state.

To further characterize the dynamics of neural activity, we compared EEG power bands between the three states. We extracted one power band per each of the 10 s non-overlapping windows per channel and averaged across these values for comparison. The powers were calculated as the area under the curve of the power spectrum for theta (1–4 Hz), delta (4–7 Hz), alpha (8–12 Hz), beta (13–30 Hz), gamma (30–40 Hz) bands and broadband power (1–40 Hz). The results are presented in Supplementary Figs. 29 and 30. In short, we observed higher power in resting state compared to self and non-self tasks in all powers, though the effect size rest versus other comparison for gamma band was rather low (theta, rest vs self: $z = 11.53$, $p < 0.001$, $r = -0.26$; rest vs other: $z = 9.11$, $p < 0.001$, $r = -0.2$;

delta: rest vs self: $z = 14.06$, $p < 0.001$, $r = -0.31$; rest vs other: $z = 12.45$, $p < 0.001$, $r = -0.28$; alpha: rest vs self: $z = 14.10$, $p < 0.001$, $r = -0.31$, rest vs other: $z = 9.71$, $p < 0.001$, $r = -0.22$; beta: rest vs self: $z = 9.88$, $p < 0.001$, $r = -0.22$; rest vs other: $z = 7.82$, $p < 0.001$, $r = -0.17$; gamma: rest vs self: $z = 5.69$, $p < 0.001$, $r = -0.13$, rest vs other: $z = 3.79$, $p < 0.001$, $r = -0.08$; broadband: rest vs self: $z = 13.43$, $p < 0.001$, $r = -0.30$, rest vs other: $z = 10.25$, $p < 0.001$, $r = -0.23$). Regarding self vs other comparisons, we observed statistically significant higher power in other condition compared to rest in every frequency band, but effect sizes are lower than 0.1 (theta: $z = 2.9$, $p = 0.004$, $r = 0.06$; delta: $z = 2.08$, $p = 0.038$, $r = 0.05$; alpha: $z = 3.54$, $p < 0.001$, $r = 0.08$; beta: $z = 2.14$, $p = 0.032$, $r = 0.05$; gamma: $z = 2.02$, $p = 0.044$, $r = 0.04$; broadband: $z = 3.38$, $p < 0.001$, $r = 0.08$).

As stated above for mice, an argument can be made against the relationship between INT and behavioral states by asserting that the relationship is spurious, confounded by neural oscillations since neural oscillations were also found to reflect behavioral states (Iemi et al.[47]; Seeber et al.[66]; Liuzzi et al.[67]; Andalman et al.[68]). To counter the argument of spurious relationship, we repeat the analyses for the mice: we first investigate the correlations between ACW and neural oscillations (theta, delta, alpha, beta, gamma and broadband) and then condition the relationship between ACW and these oscillations by adding them into the logistic regression between ACW and behavioral state. If the regression coefficient is significantly different, it is possible that the relationship is spurious. The results for correlation analyses are shown in Supplementary Figs. 31. Briefly, we observe a weak and positive correlation between INT and theta oscillatory power ($\rho = 0.069$, $p < 0.001$)

**Fig. 6 | Classification learning for EEG states.**
**a** The distribution of $\tau$ values averaged across time windows and subjects on the scalp for rest, self-narrative and non-self (other) narrative. **b** We trained support vector machines to classify rest versus self versus other states using $\tau$ values across the brain using nested cross validation for hyper-parameter tuning (10 inner, 10 outer folds). **b** Shows the confusion matrix for test data, aggregated across 10 folds. **c** Receiver operating characteristic curve for the test data of the trained support vector machine. The ROC curves for 10 folds were averaged and the standard deviation across folds was indicated with the shadings. Abbreviations: TPR True positive rate, FNR False negative rate, AUC Area under the curve, M Mean, STD Standard deviation.

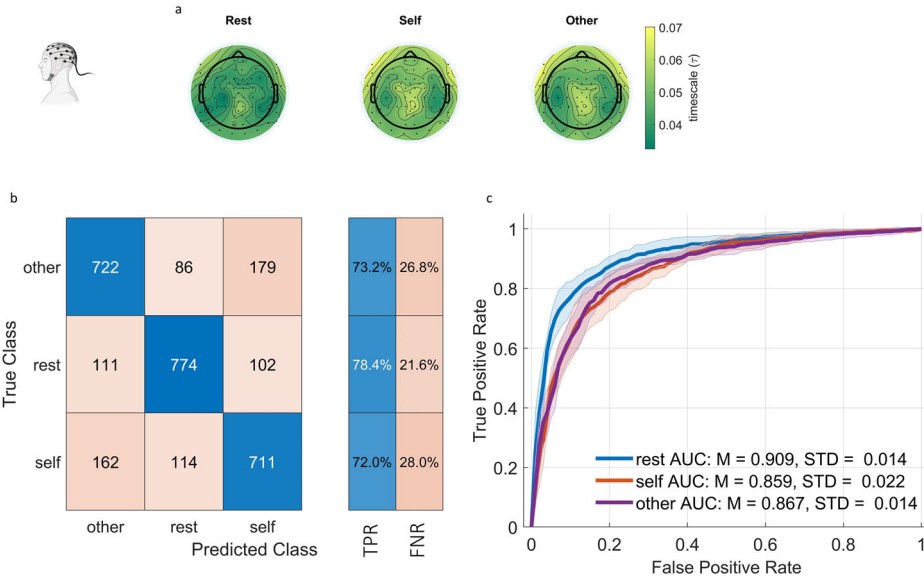

and negative correlations between INT and other power bands (delta: $\rho = -0.181$, $p < 0.001$; alpha: $\rho = -0.494$, $p < 0.001$; beta: $\rho = -0.415$, $p < 0.001$; gamma: $\rho = -0.262$, $p < 0.001$; broadband: $\rho = -0.359$, $p < 0.001$). The results for logistic regressions are presented in Supplementary Tables 3 and 4. We did not observe a significant change in the regression coefficient when it was conditioned on oscillatory powers ($z = 0.724$, $p = 0.469$). These results show that the relationship of INT and behavior does not solely depend on these additional measures of oscillatory dynamics, that is, that the INT-behavior relationship is not spurious.

As in the mice, we also looked at the rest variability of INT and rest-behavior change of the mean of INT. Fig. 7c shows resting state variability of $\tau$ values as well as rest-behavior changes of the mean of $\tau$s.

We proceeded with the correlation between resting variability and rest-task change of $\tau$s. Prior to analysis, we identified outlier data points as data points that are 3 scaled median absolute deviations away from the median and performed the analysis on data without outliers. In Fig. 7d, we, as in the mice, see a negative correlation between resting variability and rest-task percent change of $\tau$ values (rest-self difference: Spearman's $\rho(1271) = -0.25$, $p < 0.001$, rest-other difference: Spearman's $\rho(1236) = -0.32$, $p < 0.001$). The data with the outliers can be seen in Supplementary Fig. 27 (rest-self difference: Spearman's $\rho(1342) = -0.31$, $p < 0.001$; rest-other difference: Spearman's $\rho(1342) = -0.4$, $p < 0.001$). These results indicate that the variability of the $\tau$ negatively modulates the rest-task change of $\tau$ in humans just as we observed in the mice. These results for ACW-0 instead of $\tau$ are presented in Supplementary Fig. 28, showing higher ACW-0 in tasks compared to rest and negative rest variability—rest-task percent difference correlation.

Additionally, we performed the same analyses in left and right central, frontal, parietal, temporal and occipital electrodes. Supplementary Fig. 32 shows the classification of electrodes. Supplementary Fig. 33 shows rest-task differences in these electrode groups. In particular, we observed significantly higher $\tau$ values in self condition compared to rest in left central, left and right frontal, left parietal, left and right temporal electrode groups. In rest-other comparisons, left central, left and right frontal, and left temporal electrode groups show higher task $\tau$ compared to rest.

## Neural mass modeling of rest-stimulated state of INT
We used a neural mass model[78] to model our empirical findings. This model consists of 360 regions in the brain, coupled through a connectivity matrix obtained from human DTI data[65] (Fig. 8a, b). We numerically solved the model equations without and with constant external inputs to simulate resting and behavioral states respectively. Fig. 8c shows the firing rates of every region in rest (top) and active (bottom) states in one simulation. We

compared the $\tau$ values, calculated in a sliding window method using 10 s windows with no overlap (same as human EEG data). We found that in the stimulated state, the $\tau$ values are significantly higher than in the non-stimulated state (Fig. 8d, $n = 10800$ for each group, $z = 103.57$, $p < 0.001$, $r = 0.81$) mirroring the empirical findings. Moreover, as in both mice and humans, the resting state variability of $\tau$ values showed a negative correlation with the rest-stimulated state percent change of $\tau$ (Fig. 8e, $\rho(10798) = -0.420$, $p < 0.001$). Qualitatively same results for ACW-0 can be found in Supplementary Fig. 34.

## Recurrent Connections mediate rest-stimulus change of INT in the model
To test the effect of recurrent connections on INT change from rest to stimulated states, we changed the values in the diagonal of the connectivity matrix (see methods) from 1 to different values (from 0 to 4 in steps of 0.5, Fig. 9a). We compared the $\tau$ values in rest and stimulated states using Wilcoxon test. Fig. 9b shows that for recurrent connections that are lower than the default strength of connection (1), the rest-stimulated state change of $\tau$ is lower with negligible difference in the absence of any recurrent connections ($n = 10800$ for each group in all comparisons; $z = 58.71$, $p < 0.001$, $r = 0.46$ for $W_{ii} = 0.5$ and $z = 11.12$, $p < 0.001$, $r = 0.09$ for $W_{ii} = 0$). For higher recurrent connections, the difference initially increases, then stops ($z = 124.65$, $p < 0.001$, $r = 0.98$ for $W_{ii} = 1.5$; $z = 127.24$, $p < 0.001$, $r = 1$ for $W_{ii} = 2$). Increasing the recurrent connections further causes a decrease in the rest-stimulated state change with negligible difference at a high recurrent connection strength of 3 ($z = 116.67$, $p < 0.001$, $r = 0.92$ for $W_{ii} = 2.5$ and $z = 7.22$, $p < 0.001$, $r = 0.06$ for $W_{ii} = 3$). Increasing the recurrent connection strength even further causes a decrease in the $\tau$ ($z = 111.32$, $p < 0.001$, $r = -0.87$ for $W_{ii} = 3.5$; $z = 127.16$, $p < 0.001$, $r = -1$ for $W_{ii} = 4$). These results are summarized in Fig. 9c. We plotted the changes averaged across ROIs and simulations and interpolated the values between our simulations to obtain the continuous line in Fig. 9b. Qualitatively same results for ACW-0 are presented in Supplementary Fig. 35. Together, the findings of initial INT increase and subsequent INT decrease suggest that there is an optimal range in the strength of recurrent connections for modulating rest-stimulated state changes.

It can be argued that this relationship between recurrent connections and $\tau$ change is driven by differences in firing rates of the regions rather than by their intraregional recurrent connections. Indeed, when all else is held the same, the average firing rates increase with increasing recurrent connections (Supplementary Figs. 36 and 37). To counter this argument, we fixed the external input for all variations in recurrent connections so that all models show an average (across time and ROIs) firing rate of 0.1 at rest and 0.6 at

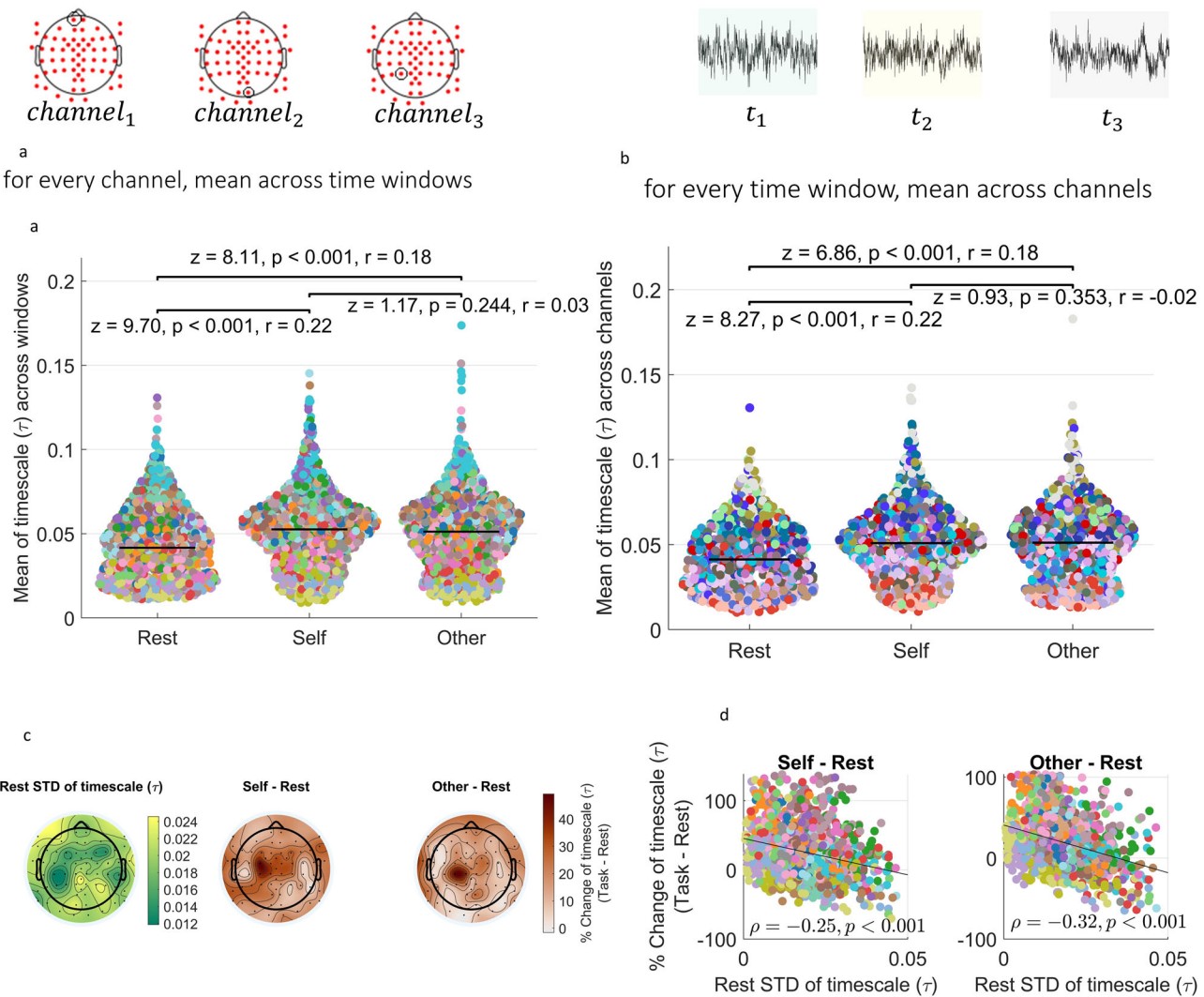

**Fig. 7 | Replication of results in the human EEG dataset. a** Comparison of τs in human EEG between three states, rest and two task states, averaged across time windows for every channel. Every dot denotes one channel. Colors denote subjects. **b** Comparison of τs across three states, averaged across channels for every time window. Every dot denotes one time window. Colors denote subjects. **c** Resting state variability and rest-task change of τs averaged across time windows and subjects. **d** We calculated the variability across time windows for each channel during the resting state and correlated that with the percent rest-task change of τ. Outliers were removed prior to the analysis. Each dot denotes one channel, colors denote subjects.

stimulation values of $W_{ii}$ from 0 to 3 (see methods). We determined the values for external input by solving the model equations for the external input in a steady state where all firing rates are equal to the targeted firing rate. Supplementary Figs. 38 and 39 show the time series of firing rates of one simulation using the determined external inputs. We ran the simulations again and compared τ values of rest and stimulated conditions. The results can be seen in Supplementary Fig. 40. In summary, we again observed a steady increase of stimulated state τ with increasing values of recurrent connections. Resting τ values on the other hand remained consistent. These findings suggest that our findings on the recurrent connections are not confounded by firing rates.

There is ample research that shows chaos in recurrent neural networks, and it can exhibit chaoticity which is a potential influencer on the timescales that we estimate[79–83]. To disentangle the potential relationship between INT and chaos, we calculated the Lyapunov spectra of our model in each configuration of recurrent connectivity and input strength. The Lyapunov spectra are shown in Supplementary Fig. 43. None of the Lyapunov exponents in any configuration were positive, indicating that our model does not show chaotic behavior making it rather unlikely that chaos is a potential confounder in our analysis.

Finally, we also tested the effect of network topology on the change of INT. Given the unique characteristics of the brain's connectivity[84–86], the complex network topology can be thought to relate to the dynamics of τ change. To test this hypothesis, we randomly shuffled the edges of the connectome while keeping the recurrent connections intact[84,85]. We performed several forms of shuffling with varying degrees of randomness from 20% of the edges to all of them in steps of 20%. The rest-stimulated state comparisons are shown in Supplementary Fig. 44. In all simulations, there was a significant change in τ values from rest to stimulated state, further cementing the role of recurrent connections in mediating the rest to stimulated state changes of τ. In sum, our modeling results show that the firing rate equations can model the empirical phenomena we observed and that recurrent connections are required for the flexible changes of τ from rest to stimulated states in this model. Moreover, given that at extreme values of recurrent connections τ start to decrease, we assume an optimal intermediate range of τ for mediating rest-stimulated state changes.

## Discussion

In this paper, through analyzing both mice calcium imaging[59] and human EEG[60] datasets, we showed that (a) distinct behavioral states can be

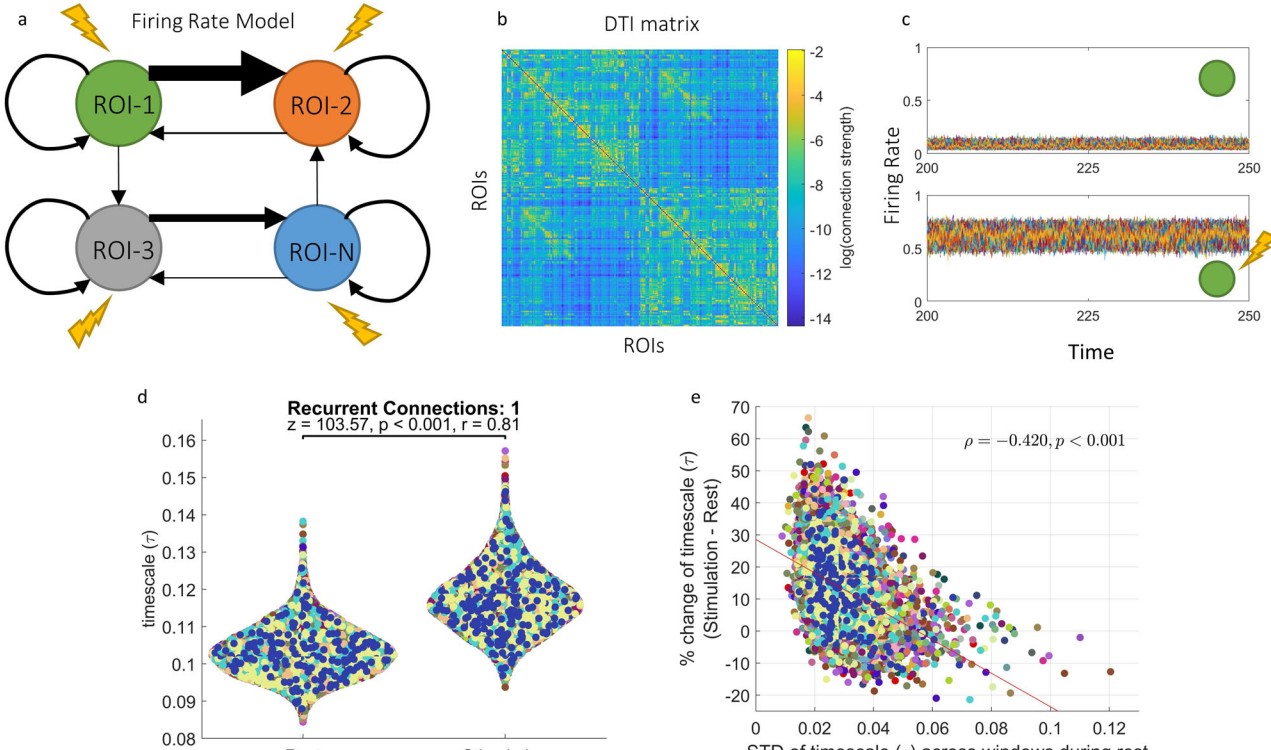

**Fig. 8 | Replication of empirical results in a firing rate model. a** We used a neural mass model to simulate the firing rates for every region in the brain. The model consists of coupled differential equations where each equation describes the dynamics of firing rate of one region. The regions are connected via a coupling matrix and have recurrent connections with themselves. **b** The connectivity matrix used in the model was obtained from human diffusion tensor imaging data. **c** An example time series for resting (top) and stimulated (bottom) conditions in the

model. Each color denotes one region. **d** We simulate the model for 300 s, discard the first 100 s and calculate $\tau$ values in 10 s sliding windows with no overlap in both rest and stimulated states. For every region, we averaged the $\tau$ values across time and compared between rest and stimulated conditions. **e** We calculated the variability of ACW-0 values across time for every region and correlated it with the percent change of ACW-0 from rest to stimulated state. In (**d**, **e**), each dot represents one region. Colors denote simulations.

predicted by their INT; (b) INT become longer from rest to a continuous behavioral state; (c) the behaviorally related change in the INT is negatively correlated with the INT resting state variability (d) neural mass modeling shows that the disruption of the recurrent connections eliminates rest-task changes of the INT. Together, our findings demonstrate the flexible nature of the brain's INT in reflecting rest-task relationships during distinct continuous behavioral states through the possible mediation by recurrent connections.

## Behavioral states reflect INT dynamics

Our first research aim was to show that behavioral states can be predicted by INT across the whole brain. Using machine learning and estimating INT via the decay rate of autocorrelation function $(\frac{1}{\tau})$, we reached high classification accuracies in predicting behavioral states using whole-brain $\tau$ in spontaneously behaving mice and humans that react to an external stimulus according to their internal states. In both cases, $\tau$ increased from rest to a behavioral state. This phenomenon is shown for both within each brain region and across brain regions.

The finding of prolonged INT in behavioral states is in accordance with the literature showing INT increases during behavioral tasks that require attention[14–17]. In particular, Manea et al.[17] showed that in time periods where spontaneously behaving monkeys engage in lever presses for reward, their INT increase compared to their immobile state. In both our analysis of the mice data and the study of Manea et al. there is no external temporal structure imposed by the experimental design. The lack of a predetermined external task structure enables the organism to generate purely spontaneous (i.e., internally initiated) and continuous behavior, limited only by the biophysical constraints of the organism.

Similarly so for the human task in our study. The task design for the human subjects also lacked a pre-determined external temporal structure except for any temporal structure in the narrative they were listening to. Specifically, subjects were required to actively track the continuous stimulus with a mouse cursor according to their internal states. The active tracking requires integration of information over longer timescales, which reflects our findings of longer timescales during the narrative listening compared to rest. Such prolongation of the timescales is in agreement with studies showing that tasks that require heightened attentional focus elicit INT increase[15,16]. Zeraati et al. explained the increase of INT by a neurophysiological mechanism involving acetylcholine, which increases vertical neural interactions while decreasing horizontal neural interactions. This serves as an example of how INT is altered both by external input, e.g., a task with stimuli, and by neurophysiological changes.

We should note that our claim about the relationship between INT and behavioral states are not specific and exclusive to INT. Previous research have shown that neural oscillations[47,66–68] and variability[69–74] also reflect behavioral states. In our supplementary analyses, we have shown that the relationship between INT and behavioral states is not spurious and confounded by variance (in mice) or oscillatory power (in human data). These results are in accordance with our claim that INT incorporates some specific information about behavioral states which can't be completely explained by or reduced to the neural activity's variance or oscillatory power.

Before going further, we should make clear the similar and different aspects of the tasks for mice and humans. The exact task details and their psychological requirements are rather distinct in mice and humans: a spontaneous behavioral motion task for mice and a self-evaluation task for humans. On a deeper level, there is a temporal similarity consisting of that

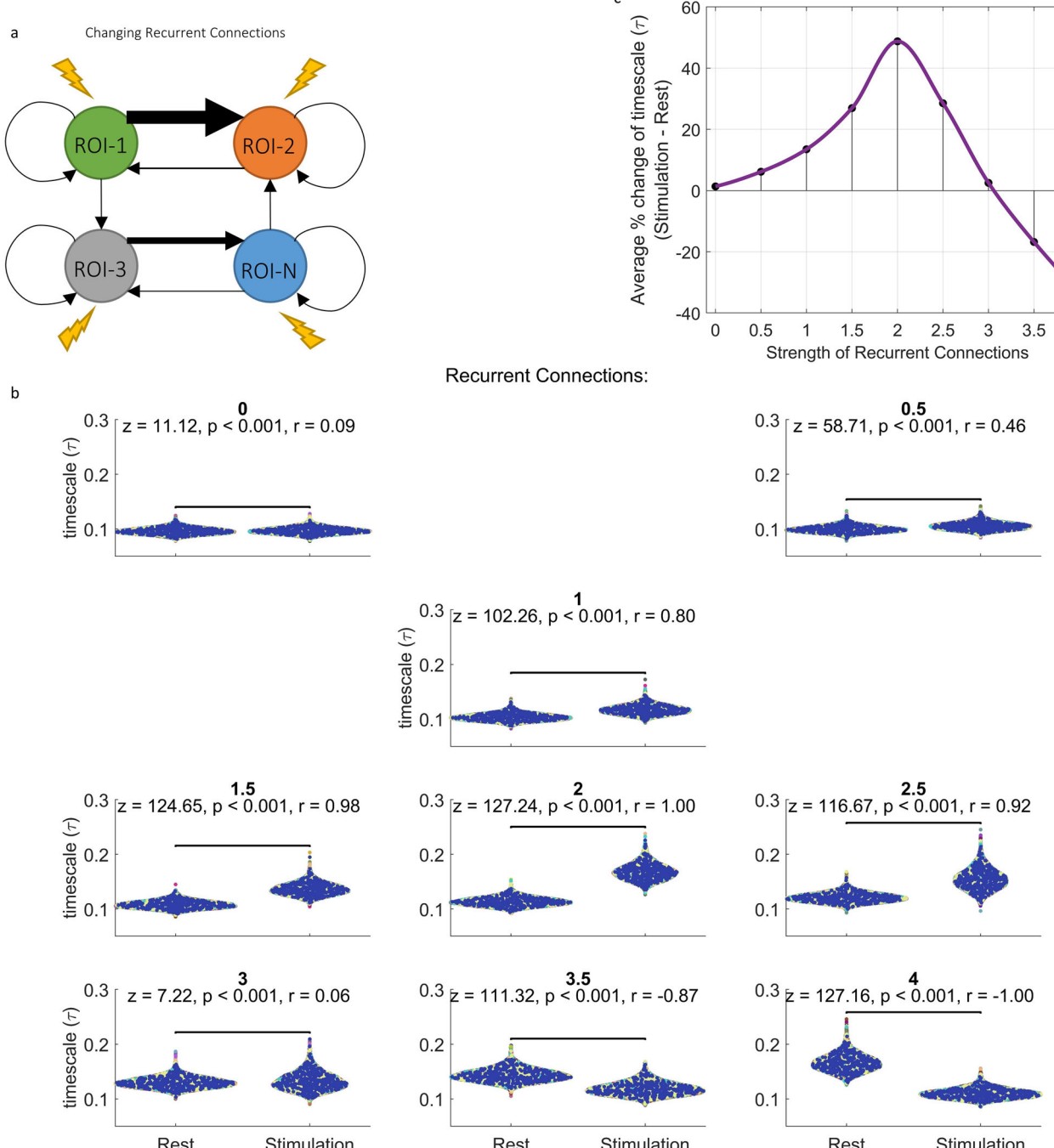

**Fig. 9 | Recurrent connections determine rest-stimulation $\tau$ change in the firing rate model. a** We change the recurrent connections in the model to test the effects of different recurrent connections on the timescale change. **b** We simulate the models with various strengths of recurrent connections and compare the timescale values in rest and stimulated states. Each dot denote one region and colors denote simulations.

**c** Relationship between recurrent connections and rest-stimulation change of timescale. The positions of the black dots are calculated as the percent change of ACW-0 averaged across regions and simulations. Continuous line was interpolated from the dots.

both tasks are continuously ongoing and thus non-interrupted during which time course they yield distinct behavioral states (like the distinct behavioral periods of the mice and the distinct varying ratings in humans). We are aware that our use of the word "behavioral" can be criticized here if one constrains the definition of behavior to the behavior of whole body, as in mice which would exclude to conceive the cursor movement in the human task as a behavioral state. However, classifying the cursor movement as a behavior is consistent with the usage of the term behavior in the literature (see for example Fig. 1 of ref. 87 and Tables 2 and 3 of ref. 88). Together, we suggest that the temporal continuity in the behavior of both mice and

human tasks can be captured by the temporal continuity of their underlying neuronal activity as measured by the INT that are based on the auto-correlation function.

We also note that the change in INT during locomotion in behaving mice was global and thus brain-wide. Only one of the mice (cm124) showed decreases in INT in a few regions. All the other mice showed increases across the whole brain in various degrees.

Here, we should note that while we describe a relationship between INT and its specificity to behavior, we do not make any assertion about either causality let alone about the directionality of this relationship. As

shown by several empirical neuroimaging studies, resting state INT can be related to cognition in various tasks[9,14,16,62,89]. However, the causal nature of this relationship and its directionality, i.e., from cognitive task to INT or from INT to cognitive task cannot be asserted based solely on our analysis of their statistical relations. It is feasible to assume that the underlying structural properties of neurons in the cortex constrain the INT which, in turn, are more fine-tuned during specific tasks[15,90].

## Negative rest-task modulation of INT dynamics

Extensive literature on single-unit cellular[49–51], EEG[41–43] and fMRI[28–36] studies show that the variability and amplitude of task/poststimulus activity is shaped by neural activity in the resting state/prestimulus period[33,36,44,51]. Motivated by these results of rest-task modulation of the amplitude and variability of neuronal activity, we investigated the relationship between the resting variability of INT and their change from rest to behavioral states. We found that a higher variability of INT during rest predicts a lower change from rest to locomotion.

The phenomenon is somewhat similar in the human EEG data where, up to a certain level of variability, the INT increase from rest to task becomes smaller. In both mice and humans, the variability of INT in rest is negatively correlated with rest to task change, e.g., behavioral change of the INT. Such negative rest-task modulation of INT extends the discussion of rest-task modulation of the amplitude to the context of timescales and behavior. Moreover, for the first time it shows the importance of the flexibility of the INT, that is, the spontaneous activity's repertoire of INT, operationalized via its variability, for behavioral states as recently hypothesized by ref. 4.

## Recurrent connections are required for the flexible rest task modulation of INT in a model of INT change

We used a firing-rate model with a connectivity matrix derived from healthy human DTI recordings[65]. When stimulated with a constant input, this model showed INT increase compared to its resting state and importantly, this increase was negatively correlated with the resting state variability of INT (Fig. 8) which replicates our findings in both mice and human data. One may nevertheless raise the question whether our model and its stimulation really reflect the rest and behavioral states in empirical data.

The stimulation in the model can be thought of as external stimulus in EEG data or it can also correspond to an increase in the excitability as in increased arousal of mice estimated by the pupil size. These two statements are mathematically equivalent in the formulation of the firing rate equations since the bias term determines excitability and it is linearly added on external input. Given that both forms of stimulation in the empirical data sets, i.e., external stimulus and internal excitability, are identical in the model equations, we assume that our model mirrors the states in both mice and humans.

We also observed that the change in INT requires the existence of recurrent connections. In a previous study, Zeraati et al.[16] showed that the flexibility of timescales can be accounted for by small changes in recurrent connections in a locally connected lattice network where every node can have two states as long as the network is poised near criticality. Here, we extend these observations of recurrent connections to a network model with biophysically realistic connectome at the whole brain level showing that the flexibility of timescales under stimulation is possible only with sufficiently strong recurrent connections. Moreover, the relevance of the intraregional recurrent connections is further underlined by our observation that changes in the topological structure or interareal connection strengths of the connectivity network did not lead to changes in the INT from rest to stimulated state.

Moreover, our modeling indicates an optimal regime for the level of recurrent connections. Increasing the recurrent connections increases the efficacy of external input and saturates the model towards very high firing rates. As a result, with very strong recurrent connections, the model shows decreased instead of increased rest-stimulated state change of INT. This indicates that there is an optimal regime where the model can realistically simulate empirical data from healthy participants. Importantly, this optimal regime seems to lie in more or less intermediate values in the strength of the recurrent connections which fares well with the assumption that the relationship between structure and dynamics follows the principle "average is good, extremes are bad"[91].

Finally, albeit tentatively, our findings carry relevance for mental disorders like schizophrenia. Schizophrenia is characterized by abnormal INT in both rest and task states[92–95]. Moreover, the INT show reduced change from rest to task in schizophrenia[92] (see also[96] for a review of reduced rest-task changes in schizophrenia). May such reduced rest-task change be related to changes in recurrent connections in the psychotic brain? It was recently hypothesized that altered synaptic pruning during adolescence can lead to decreased recurrent connectivity especially in the prefrontal cortex; which makes the brain vulnerable to psychosis[97–99]. Future empirical and theoretical studies are required to link these questions to the dynamics of INT including their potentially underlying recurrent connections.

## Limitations

A few limitations need to be mentioned before we conclude the article. Firstly, even though our EEG results showed some localized results, we lack individual MRI scans of subjects and therefore did not perform any source localization. Secondly, we refrain from making any causal or mechanistic statements based on our modeling. It should be mentioned that the causal connection between recurrent connections and INT change is only mathematical and not biological. Further research needs to be performed to assess this hypothesis thoroughly. Another limitation are possible confounding factors in the EEG task. The continuous movement of the mouse cursor might induce excessive motion artifacts. A higher head motion during any one of the tasks (self or non-self versus resting state) can influence INTs and alter the results. While we used standard well established methods for artifact rejection[100–105] to eliminate non-physiological noise from the data via artifact rejection, we cannot fully exclude that motion artifacts may still confound our EEG results. Yet another limitation of the paper is a lack of behavioral measures reflecting arousal in human data. The addition of arousal would strengthen the arguments about behavioral relevance. The relationship between arousal and INT in task states can be a future research avenue, to be explored further.

We should note that while our research did not incorporate the notion of chaos explicitly, there is ample evidence of chaos in the dynamics of recurrent neural networks[79]. Research has shown that chaos, and more specifically edge of chaos increases the ability of recurrent neural networks to learn to generate complex periodic patterns of activity from input[80] and represent timing[81,82]. Furthermore, it was shown that external inputs can suppress these chaotic dynamics[83]. Our future research direction will involve harnessing the chaotic dynamics of recurrent neural networks for the spontaneous generation of behavioral states and how the chaoticity of neural networks impact INT.

## Conclusion

Biological organisms show highly flexible behavior along different timescales. How this flexibility of continuous behavior relate to the brain's INT is an open question. We here demonstrate that the flexibility and thus repertoire of INT operationalized by the variability of INT during rest modulates the changes in INT during behavior in both mice and humans. We found that INT across the brain can predict behavioral states during which the INT become longer relative to rest. The variability of INT in the resting state was found to correlate negatively with the rest task changes of the INT. Finally, our findings were replicated in a neural mass model which allowed us to show that recurrent connections are necessary for the flexible change of INT from rest to task states. Our results support and add to the previous findings on the behavioral relevance of INTs by showing their dynamic changes during distinct continuous behavioral states as possibly mediated by recurrent connections.

## Methods

### Data acquisition and preprocessing

**Calcium imaging.** We used the publicly available preprocessed wide-field optical mapping (WFOM) data from ref. 59. The details of data acquisition and preprocessing can be found in ref. 59. In brief, neural activity of five age-matched spontaneously behaving male Thy1-jRGECO1a transgenic mice (expressing red-shifted calcium indicators) were recorded. Each mouse was scanned up to eight sessions. Each session consists of up to 10 10-min recordings. The sampling rate of data is 20 Hz. In addition to neural activity, behavioral activity was also recorded to track pupil size and whisker movements of the mice as well as the velocity of the wheel they stand on. JRGECO fluorescence recordings were corrected for hemodynamic cross-talk and converted into $\Delta F/F$. A high-pass filter at 2 Hz was applied and the data was further denoised with principal component analysis. To determine regions of interest, the authors used k-means clustering and picked the number of clusters according to a non-negative least squares fitting procedure that generates spatial representations of data as coefficients for time courses of each ROI. The number that gives the best fit to this analysis was picked as the optimal number of clusters (92 regions). Time courses were extracted as the average activity inside the ROIs. All the experimental protocols on mice were reviewed and approved by the Institutional Animal Care and Use Committee at Columbia University (Protocol Number: AC-AAAS3453). We have complied with all relevant ethical regulations for animal use.

**Human EEG.** We analyzed a dataset that was collected for a previous study using a modified version of self-evaluation task[60,77]. The exact details of acquisition and preprocessing is available at ref. 60. In brief, 27 (10 females, mean age = 30.3 years) healthy subjects from the local community in Ottawa, Canada were scanned. The scanning procedure involved an 8-min eyes open resting state while staring at a fixation cross and a task paradigm adapted from Vallacher et al.[77]. The subjects recorded an 8-min autobiographical narrative and this narrative was presented to them during the experiment. The subjects were instructed to continuously evaluate the narrative's contents as positive or negative using a mouse cursor. Keeping the cursor close to the center indicated a negative content whereas away from center indicated positive, thus enabling them to track their internal state based on the narrative. The audios were presented with headphones. For ease of distinction, the cursor was colored green in the proximity of the center, red for away from center and yellow for intermediate distances. Additionally, all participants verbally confirmed their understanding of the cursor task instructions before beginning the experiment. This constituted the "self" condition of the task. For non-self (other) condition, a member of our research team recorded an 8-min long narrative. The non-self narrative was also presented to the same subjects and they asked to make the cursor movements for their emotional states. Cursor movements were recorded with 200 Hz sampling rate whereas EEG was recorded at 1000 Hz. Preprocessing of EEG data involved (1) downsampling to 500 Hz, (2) bandpass filtering between 0.5 and 50 Hz, (3) spherical interpolation of channels that were flat for at least 5 s or channels that had amplitude exceeding three interquartile ranges, (4) re-referencing of channels to average, (5) independent component analysis of 62 components and rejection of non-neural components using multiple artifact rejection algorithm (MARA[100]). In the end, four participants were excluded from EEG analysis due to the noisy channels, one participant was excluded due to the absence of alpha peak and finally one more subject was excluded due to technical issues relating to data acquisition. Due to discrepancies in subjects' recording length, both self and non-self recordings, as well as resting state were truncated to 470 s which is the length of the shortest recording. It needs to be mentioned that while we tried our best to eliminate any noise unrelated to brain activity during preprocessing via artifact rejection using standard and well established methods (which[100] was used by ref. 101–105 among many others), we nevertheless cannot completely exclude motion artifacts as a possible confounder. The experiments were approved by the local ethics board (REB NUMBER: 2016004) of the University of Ottawa Institute of Mental Health Research and informed consent was obtained from all participants. All ethical regulations relevant to human research participants were followed.

### Determination of behavioral states in mice

We used the same behavioral classification used in ref. 59. This gives 5 distinct behavioral states: onset of locomotion (onset): overlapping 5 s rest and 5 s running; locomotion (running): the running periods with at least 20 s locomotion duration and 60 s pre-locomotion rest (10 s in the middle were picked for analysis); locomotion offset (offset): 5 s at the end of locomotion and 5 s at the beginning of rest for locomotion bouts that last at least 20 s and have at least 10 s post-locomotion rest; initial rest: first 10 s immediately after the end of locomotion for locomotion bouts that last at least 5 s with at least 60 s post-locomotion rest; sustained rest: 10 s of rest starting 40 s after the end of locomotion state for the same locomotion bouts as for the initial rest state. All the analyses were performed on these 10 s windows.

### Estimation of intrinsic neural timescales

Autocorrelation function was used to estimate INT for behavioral states. The autocorrelation function $r$ of a signal $x$ is the signal's correlation with itself on different lags:

$$r_l = \frac{\sum_{t=1}^{N-l}(x_t - \bar{x})(x_{t+l} - \bar{x})}{\sum_{t=1}^{N}(x_t - \bar{x})(x_t - \bar{x})} \quad (1)$$

where $l$ denotes lag, $\bar{x}$ denotes the mean of $x$ and $N$ is the number of sampling points. We estimated the INTs by fitting an exponential decay function $e^{-\frac{1}{\tau}l}$ and extracted the parameter $\tau$[8,106,107]. $\tau$ corresponds to how long does it take for the autocorrelation function to reach the value $\frac{1}{e}$.

We replicated our main results with another estimation ACW-0 which is defined as the lag where autocorrelation function reaches 0[9,21]. The results for ACW-0 can be found in Supplementary Material, Figs. 4–6, 10–12, 18, 21–23, 28, 34, and 35.

In the calcium imaging data, we calculated one $\tau$ value per behavioral segment and in EEG data, similarly we calculated one $\tau$ value per each 10 s window of data with no overlap between windows.

### Classification of behavioral states using support vector machines

In order to show that behavioral states can be distinguished based on the topography of $\tau$, we used SVM with a radial basis function as kernel. The data was organized as all the time windows (for mice, $n = 822$; for human EEG, $n = 2961$) as samples and the $\tau$ values for each ROI (92 ROIs in total for mice, 64 channels for human EEG) as features. For hyperparameter tuning, we used nested cross validation with 10 inner and 10 outer folds to avoid data leakage. The hyperparameters regularization strength lambda and kernel scale parameter were tuned using MATLAB's bayesopt function with default settings. We report the confusion matrix and one-versus-all receiver operating characteristic (ROC) curve calculated on test data. An ROC curve shows the true positive rate as a function of false positive rate for different cut-off points. The area under an ROC curve (AUC) indicates how much a model can distinguish various groups of data. For the confusion matrices, we aggregated the results from 10 outer folds. Since the sampling was done without replacement, aggregated test data of 10 outer folds correspond to whole dataset. For the ROC curves in the manuscript, we averaged the ROC curves of each fold and indicated their SD as shading around the mean. The confusion matrices and ROC curves for each fold can be found in Supplementary Material (Supplementary Figs. 2, 3, 19, 20).

While SVM is a powerful tool for classification, the nonlinear transformation it performs on the data using a radial basis kernel makes the results more difficult to interpret. In addition to SVM, we performed logistic regression as well. Logistic regression models the probabilities of each

behavioral state as a linear combination of predictors, therefore it is easier to interpret. The data is organized as the same way as SVM. As in SVM, we used nested cross—validation with 10 inner and 10 outer folds to tune the hyperparameter lambda, denoting regularization strength. We set the relative coefficient tolerance for terminating the optimization to 0.0001. We report confusion matrix and one-versus—all ROC curve calculated on test data and aggregate results from 10 outer folds for confusion matrix, this is the same as in SVM. The averaged ROC curves and aggregate confusion matrix can be found in Supplementary Figs. 7, 24. The ROC curves and confusion matrices for each fold can be found in Supplementary Figs. 8, 9, 25, 26.

### Additional analyses to further investigate neural dynamics

To characterize the neural activity more thoroughly, we compared a number of features between the behavioral states in mice and human data. In mice data, we compared the mean and SD of the calcium imaging activity between sustained rest and locomotion states. In human EEG data, we compared the power bands (theta: 1–4 Hz, delta: 4–7 Hz, alpha: 8–12 Hz, beta: 13–30 Hz, gamma: 30–40 Hz, and broadband: 1–40 Hz) across the three states (rest, self, non-self). The mean and SD from mice and the power bands from humans were calculated in the same sliding window fashion as the other measures: we extract one measure per ROI/channel from each 10 s non-overlapping time window. To extract power bands, we used the periodogram method and calculated the area under the curve of the power spectrum using the trapezoid method.

### Statistics and reproducibility

We used Wilcoxon tests for pairwise comparisons. All the $p$ values are for two-sided statistical tests. $p$ values were corrected for multiple comparisons using Bonferroni-Holmes method. We estimated the effect sizes for Wilcoxon tests using biserial correlation r. Mixed effect models were implemented using JASP using Satterthwaite method for testing the significance of model terms[108]. Sample sizes for each analysis are given in the main text.

### Biophysical modeling

We used a firing rate model that consists only of excitatory connections to model our findings. The choice of excitatory connections ensures simplicity in the model, avoiding parameters related to inhibition. In addition, we replicated the main result of our paper in a model that also includes inhibitory connections. This is achieved via incorporating one inhibitory and one excitatory population to each region. The model specification and results can be seen in supplementary material section "Model with inhibitory connections". The model we used for the main manuscript is the following:

$$\tau_i \frac{dx_i}{dt} = -x_i + f\left(\sum_{j=1}^{N} W_{ij}x_j + b + s + I\right) \quad (2)$$

$$f(x) = \frac{1}{1 + e^{-rx}} \quad (3)$$

Where $x_i$ is the firing rate of i-th region, $b$ is the bias term, $s$ is 0-mean unit variance gaussian noise, $I$ is external input and $f(x)$ is the sigmoid input-output transfer function that was also used in Wilson–Cowan equations[78]. We set $b = -3$, $r = 0.5$. For resting state, we set $I$ as 0 whereas for stimulated state, we set $I = 1$. $W$ is the connectivity matrix that was taken from averaged healthy human DTI data[65] which is based on the HCP-MMP 1.0 atlas[109] and includes 360 regions of interest. The DTI matrix consists only of interareal (long-range) connections, leaving the intra-areal/recurrent connections unspecified. The off-diagonal elements of $W$ are, therefore constrained by the DTI matrix. The diagonal elements determine recurrent connections. We set them as 1 for the "default" state of the network and changed them to explore the effect of recurrent connections on timescales. Finally, following the findings that report the sum of interareal connections are stronger than

recurrent connections[110–113], we scaled the off-diagonal elements of $W$ so that on average, $\sum_{i \neq j} W_{ij} = 2$. We simulated the model 30 times for each state (changing $I$ and diagonal elements). We investigated the recurrent connections with the values from 0 to 4 with steps of 0.5. Each model was simulated using a second-order Runga-Katta method with a step size of 0.01 s for 300 s. The first 100 s were discarded to avoid the possibility of initial conditions affecting the results. The firing rates relate to calcium imaging data since the fluorescence of a cell reflects the average activity of the calcium sensors. The activity of the calcium sensors reflects the average calcium concentration over the past few hundred milliseconds. Calcium concentration finally reflects the number of spikes fired by the cell, leading us to firing rates we model via the equations described above[57,114,115].

To account for the possible confounding effect of firing rate on time-scales, we run control analyses where we set I as a function of time for each value of recurrent connections so that in the resting state, the mean of firing rate across time and regions is 0.1 and for stimulated state, it is equal to 0.6. To achieve this, we modify the equations by including dynamics for I:

$$\tau \frac{dx_i(t)}{dt} = -x_i(t) + f\left(\sum_j W_{ij}x_j(t) + b + I_i(t)\right) \quad (4)$$

$$\tau_I \frac{dI_i(t)}{dt} = \widetilde{I}_i(t) - I(t) \quad (5)$$

$\widetilde{I}_i(t)$ denotes the desired input for region $i$ to reach the desired firing rate. We denote the targeted firing rate as $\bar{x}$, which is same for every region in the model. To determine $\widetilde{I}_i(t)$, we start with noting $f(x)$ and its inverse $f^{-1}(x)$:

$$f(x) = \frac{1}{1 + e^{-rx}} \quad (6)$$

$$f^{-1}(x) = \frac{\log\left(\frac{x}{1-x}\right)}{r} \quad (7)$$

To calculate $\widetilde{I}_i(t)$, we look at the steady state solutions for $x_i$ and replace $x_i$ with $\bar{x}$ since we want $x_i = \bar{x}$ in the steady state. Additionally, we replace $I_i(t)$ with $\widetilde{I}_i(t)$ since again, we claim that $\widetilde{I}_i(t)$ is the input value that would give us $x_i = \bar{x}$ and based on (5), $I_i(t)$ will approach $\widetilde{I}_i(t)$ exponentially with the rate $\tau_I$:

$$\bar{x} = f\left(\sum_{j \neq i} W_{ij}x_j(t) + W_{ii}\bar{x} + b + \widetilde{I}_i(t)\right) \quad (8)$$

$$\frac{\log\left(\frac{\bar{x}}{1-\bar{x}}\right)}{r} = \sum_{j \neq i} W_{ij}x_j(t) + W_{ii}\bar{x} + b + \widetilde{I}_i(t) \quad (9)$$

$$\widetilde{I}_i(t) = \frac{\log\left(\frac{\bar{x}}{1-\bar{x}}\right)}{r} - \sum_{j \neq i} W_{ij}x_j(t) - W_{ii}\bar{x} - b \quad (10)$$

Supplementary Figs. 38 and 39 show the firing rates for all regions and determined values of $I$. To solve the equations numerically, we set $\tau_I$ to 0.05 and reduce the time step to 0.01.

It is important to note that RNNs can exhibit chaotic behavior which can potentially influence INTs. To eliminate the potential confounding nature of chaos, we investigated the Lyapunov spectra[116] of our model. To calculate Lyapunov exponents, we used the ChaosTools.jl package[117] which is part of the larger ecosystem of DynamicalSystems.jl. The software we used uses H2 algorithm introduced in ref. 118. Briefly, this method uses QR decomposition on D—dimensional deviation vectors of tangent dynamical system, where D is the dimensionality of the model (in our case, 360 corresponding to 360 regions of interest). The QR decomposition at each step yields the local growth rate of each dimension. This procedure is done

through N steps and the growth rates are averaged across these N steps, yielding the Lyapunov exponent for each dimension. In our case, N was chosen to be 5000, with dt = 0.01, yielding 50 s of simulation. The Lyapunov spectra for each value of recurrent connections and rest versus stimulated states are given in Supplementary Fig. 43.

Finally, we also tested the effect of network topology on the change of INT. This amounts to a randomization of structural connectivity. We randomly shuffled proportions of the edges of the connectivity matrix while keeping the diagonal elements intact. We explored various degrees of shuffling in percentages from 20 to 100, with 30 simulations for each degree of shuffling. As in the original analysis, we calculated $\tau$ values in 10 s windows with no overlap.

## Reporting summary

Further information on research design is available in the Nature Portfolio Reporting Summary linked to this article.

## Data availability

The preprocessed Calcium imaging data can be accessed at https://zenodo.org/records/7968402. The human EEG data are not publicly available due to ethical requirements and privacy concerns. The numerical data that can be used to generate figures are stored in figshare with https://doi.org/10.6084/m9.figshare.27910299.v1[119].

## Code availability

The code used for analyses is available at https://github.com/duodenum96/acw_behavior.

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

## Acknowledgements

G.N. is supported by the European Union's Horizon 2020 Framework Program for Research and Innovation under the Specific Grant Agreement no. 785907 (Human Brain Project SGA2), UMRF, uOBMRI, CIHR, and PSI. He is also grateful to CIHR, NSERC, and SSHRC for supporting the tri-council grant from the Canada-UK Artificial Intelligence (AI) Initiative "The self as agent-environment nexus: crossing disciplinary boundaries to help human selves and anticipate artificial selves" (ES/T01279X/1) (together with Karl J. Friston from the UK).

## Author contributions

Y.Ç. and G.N. conceived the project. Y.Ç. analyzed the data, created figures and performed simulations. G.N. supervised the project. D.S. provided human EEG data. Y.Ç. and G.N. wrote the first draft. K.K., A.W., P.K. commented on the manuscript and contributed to revisions.

## Competing interests

The authors declare no competing interests.
