## [Transparent Peer Review file · Communications Biology]

Flexibility of Intrinsic Neural Timescales During Distinct Behavioral States

Corresponding Author: Dr Yasir Çatal

Version 0:

Reviewer comments:

Reviewer #1

(Remarks to the Author)

Review: Intrinsic Neural Timescales Flexibly Mediate Behavioral States

This work by Catal and colleagues investigates how intrinsic neural timescales (INT) mediate task states during resting flexibly. To address this, authors use calcium imaging data from rodents during rest and locomotion as well as human EEG recorded during cognitive tasks involving self-and-non-self-related processing. Finally, they use a neural mass model to investigate the underlying mechanisms to explain empirical observations during rest and task states with computational modelling. They have reported few primary findings: (i) Behavioral states during tasks in both rodent and human data can be accurately predicted from INT, (ii) INT becomes longer during task states compared to rest, (iii) INT changes from rest to task is correlated negatively with the variability of INT during rest, (iv) neural mass modelling shows a key role of recurrent connections in mediating the rest - task change of INT.

Although the questions raised here are sufficiently interesting empirically and computationally and may benefit diverse research areas that investigate the dynamic nature of the brain's INT in shaping behaviour and brain states, there are also many areas of real concern for this work. Although this cross-species attempt to address heterogeneity and variability of neural time scales is timely from the outset, and their correlation with behavior looks exciting and interesting, fundamentally speaking, there are some major drawbacks and loopholes that the authors do not address with scientific rigour to draw overarching conclusions what they would like to reach. Below, I outline my detailed comments, major problems, and suggestions to help authors review these points carefully while revising.

Major comments

Is INT modulating stimulus- or task-related activity, or could something else (underlying biological substrate) modulate intrinsic neural time scales derived from brain signals that may constrain stimulus-driven activity in a specific way? Please consider rephrasing this in the abstract.

The authors cite recent neuroimaging work in the introduction to motivate the rationale for examining INT modulation during goal-directed behavior/task. However, I am very concerned and hesitant to accept that INT modulates various aspects of cognitive processing in myriad task scenarios. What is true is the heterogeneity & hierarchy of time scales and their association with spontaneous brain-wide activity and task-specific modulation of INT, as shown by previous primate and human studies. However, the converse is difficult to digest, stating that INT modulates goal-directed tasks or task switching. Even if we accept the example, the author cites a conversation composed of words, sentences, or perhaps higher-order stimuli such as music associated with different temporal receptive windows (TRWs) of processing specific brain regions of interest or involves distributed processing in the salient brain regions. The underlying mechanisms that modulate TRWs across sensory and cognitive processing remain elusive. The present introduction requires more clarity to situate the relevant research questions they are investigating, or the scope is too narrow. Listening to real-life conversations, auditory narratives, music, watching movies, and all naturalistic tasks involves complex processing. The only change in time-scale features alone predicts complex cognitive functions that are difficult to digest. I would rather think similarly to what was proposed by Gao et al., 2019 (the authors have cited) that INT is functionally very dynamic and constrained by underlying structural properties of neurons in the cortex, particularly during spontaneous activity, and perhaps, get more finely tuned during specific tasks.

Gao, R., van den Brink, R. L., Pfeffer, T. & Voytek, B. Neuronal timescales are functionally dynamic and shaped by cortical

microarchitecture. *eLife* 9, e61277 (2020).

Also, very similar findings here in the context of neurodevelopment and atypical neural timescales by Watanabe et al., 2019 Watanabe, T., Rees, G., & Masuda, N. (2019). Atypical intrinsic neural timescale in autism. *Elife*, 8, e42256.

A related point to the above is debatable; as the authors mentioned, the current work indicates that INT becomes longer with task engagement and increased attention during behavior, as shown recently by Zeraati, R. et al.2023 (cited by the authors) also suggests that increased attention and stimulus engagement, which modulates INT not the reverse that INT causally increases attention and stimulus engagement reinforcing what was mentioned earlier.

Author states, "How the INT in naturalistic behavioral states relates to the timescales of the brain's spontaneous activity remains yet unclear. Addressing this question is the goal of this study. However, I would like to ask what is ambiguous: what do they mean by behavioral states, and what is the difference between behavioral and naturalistic states? What are the key differences between behavioral states and brain states? There have been numerous occasions when these terms have been introduced in neuroscience to interpret different things and are very debatable, hence, an operational definition/meaning is much needed to offer any insight.

The author cites a handful of selective studies to indicate that modulation of task INT by rest INT is motivated by a wide range of studies demonstrating that the dynamics of rest or pre-stimulus state can modulate the brain's response to an external stimulus. On the contrary, I can say numerous studies show the brain's intrinsic dynamics during resting state are very different from different categories of sensory and stimulus-driven goal-directed tasks, and often, prestimulus brain states have no bearing on predicting stimulus-induced response/behavior. Consequently, giving null results. Also, the dimensionality of resting-state brain states differs greatly from task-evoked states operating on a low-dimensional subspace. Hence, I would, at best, say the evidence is mixed, and it is too hasty to postulate a connection/associate to put forward a hypothesis based on selective sampling of a few studies borrowed from primate electrophysiology, fMRI and EEG vastly operating on different physiological time scales.

If I understood the authors correctly, then they are simply trying to answer the question of whether INT is behaviourally relevant or not. In that case, the pertinent question comes how a task that is carried out by mice in spontaneous awake condition, such as their calcium imaging data recorded during spontaneous behavior, relates to human EEG data recorded from participants while they listen to autobiographical (self-related) and non-autobiographical (non-self / other related) narratives. I am not fully convinced they are related.

Secondly, any number of training features can be used to train a classifier model (they use a rather simplistic ML model like SVM). This is fine. However, the only handcrafted feature they use here is INT. Hence, method-wise, I do not see the rationale for using a classifier instead of simply using multivariate regression models based on INTs estimated empirically from brain areas (ROIs) to predict behavioral response (dependent variables). What the authors have done instead is a classification task to see whether the true labels versus labels predicted by the classifier model agree to what degree of accuracy. Suppose they already know the class labels, e.g., onset of locomotion, locomotion, the offset of locomotion, initial rest, and sustained rest. Wouldn't it be better to employ a prediction approach according to their whole-brain INT estimates rather than a classification task to classify behavioral states? I would suggest that the authors review their approach.

Any possible confounds that may arise due to the nature of the self-versus non-self-task in human participants and their implications need to be mentioned in the methods and discussions. While taking cursor-based tracking of responses, particularly valence, how the participants have been instructed during the behavioral task is fundamentally important. What ensures the arousal experience in participants experienced due to the nature of the stimuli also needs to be verified by tracking behavioral parameters associated with this. Finally, I am very concerned about how the task designed for humans has anything to do with the rodent task determining behavioral and brain states in mice. At least, the authors have yet to show any convincing data to show at some level, neural and behavioral responses by humans and rodents are congruent. They seem to be very independent behavior measures in two different species. In my opinion, what binds this cross-species investigation is still somewhat unclear.

Finally, they use the Wilson-Cowan type of firing rate models to simulate neural masses to investigate underlying mechanisms based on recurrent connectivity among neurons potentially; however, how the population firing rates relate to calcium imaging data, which measures fluorescence as a proxy of voltage-gated-ion channels in mouse data, needs to be clarified. This needs to be mentioned clearly in the methods.

I think it would be better if, instead of recursively tuning the learning parameter to reach a target excitatory firing rate, they consider using a dynamic equation that up or down-regulates the learning rate depending on the state of the neurons or whether the target rate is achieved. This, I think, theoretically speaking a more accurate formulation than the gradient descent method that was run until the error $<_x>_{time,_{rois}-target}$ was lower than the tolerance value of 0.02, then run 10 more times.

Please explicitly mention that the mathematical model you use has only one variable, which is excitatory. I do not see any inhibitory variable. Typically, when we speak of recurrent connections, it is between E-I coupling in a biophysical sense. Hence, when the authors say we use the large-scale biophysical model to investigate the role of recurrent connections on the flexible rest task changes of the INT, I am not sure which recurrent connections they are talking about. Finally, what differentiates short-range (recurrent) and long-range (typically excitatory) connections is if they use diffusion tensor imaging (DTI) or human connectome data to constrain the anatomical connectivity between relevant ROIs. Many of these things are

missing from methods, making it harder to understand and appreciate their model (despite a schematic in Figure 8).

Reviewer #2

(Remarks to the Author)

Brief summary

The authors examine how intrinsic timescales (autocorrelation) of neural activity relate to certain types of behavior. They examine calcium activity in behaving mice and EEG activity in humans performing a cognitive task, finding that INT predicts behavioral state and influences future activity in a state-dependent manner. These findings are supported by a RNN model, which shows that INT is effected by auto-connection strength in the weight matrix.

Overall impressions

The finding that INT can predict - and influence - behavior is of significant interest to the field. The authors replicate their result in two different datasets, significantly strengthening their argument. Examining the network mechanisms using a model provides additional predictions for future work for the field and makes the overall work more compelling. Additional work is needed to show the specificity of the effect, and provide additional information and context to understand the model. This work will significantly strengthen the author's arguments.

Specific comments

1. The finding that the autocorrelation, or intrinsic timescale, of regional cortical activity is very interesting. To interpret this finding it is important to characterize the neural activity itself more thoroughly, especially because correlation is influenced by magnitude of the activity. For example, what is the activation rate and variability (e.g. Fano factor) of neurons in the calcium imaging data? Are there changes in the power spectra across states in the EEG data?
2. Because the authors argue for the specific significance of INTs, it's important to demonstrate whether their findings are specific to that metric. Additional comparators are needed. For example, how well does the simple mean or variance of the calcium activity predict behavioral state in the SVM? Similarly, how well does average power predict state in the EEG data? Sub bands?
3. There is a significant amount of research on the chaotic nature of RNNs. As the concept of an intrinsic timescale boils down to the autocorrelation of the data and this the stability of the network, the authors should cite some of it. For example Dean Buonomano, Larry Abbott, and Kanaka Rajan have all investigated this issue.
4. The stability (or chaotic nature) of the RNN would impact the autocorrelation of the simulated data. The primary motivation of the simulation is to examine the network properties (e.g. connection strength) that may drive the change in autocorrelation. Without some additional characterization of the network structure and dynamics it is difficult to interpret the results of the simulation. For example, the authors could examine the eigenvalues of the network's weight matrix or measure the Lyapunov exponent of the network trajectories.
5. The authors use the model to support one their primary arguments: that the timescales of neural activity relate to - and predict - states of motor behavior. However, it is unclear whether the RNN developed in the paper could itself drive motor behavior. This would additionally impact the network's sensitivity to input strength. Developing a model that can simulate motor activity would demonstrate that it approximates cortical activity, improve its validity, and significantly strengthen the argument of the paper.

For examples see:

Sussillo, D., and Abbott, L.F. (2009). Generating coherent patterns of activity from chaotic neural networks. *Neuron* 63, 544–557. <https://doi.org/10.1016/j.neuron.2009.07.018>.

Laje, R., and Buonomano, D.V. (2013). Robust timing and motor patterns by taming chaos in recurrent neural networks. *Nat. Neurosci.* 16, 925–933. <https://doi.org/10.1038/nn.3405>.

Rajan, K., Abbott, L.F., and Sompolinsky, H. (2010). Stimulus-dependent suppression of chaos in recurrent neural networks. *Phys Rev E Stat Nonlin Soft Matter Phys* 82, 011903. <https://doi.org/10.1103/PhysRevE.82.011903>.

Hardy, N.F., Goudar, V., Romero-Sosa, J.L., and Buonomano, D.V. (2018). A model of temporal scaling correctly predicts that motor timing improves with speed. *Nature Communications* 9, 4732. <https://doi.org/10.1038/s41467-018-07161-6>.

Version 1:

Reviewer comments:

Reviewer #1

(Remarks to the Author)

I have no further comments for the authors based on my reading of their revised manuscript. In the revised version, the authors have carried out additional analysis as well revising the manuscript substantially to adequately address all my comments, concerns and questions. In my opinion, the revised version has improved and the authors were able to provide necessary amendments wherever it was applicable to improve the narrative and remove fuzziness. I think the contribution in this manuscript focusing on characterizing the flexibility of intrinsic neural time-scales during behavioural states across species makes an important advance in our understanding of neurobiology of timescales associated with behaviour and cognition.

Reviewer #2

(Remarks to the Author)

Brief summary

The authors examine how intrinsic timescales (autocorrelation) of neural activity relate to certain types of behavior. They examine calcium activity in behaving mice and EEG activity in humans performing a cognitive task, finding that INT predicts behavioral state and influences future activity in a state-dependent manner. These findings are supported by a RNN model, which shows that INT is effected by auto-connection strength in the weight matrix.

Overall impressions

The authors have added a number of new analyses which make the work more interpretable and conclusive. In addition, they have made changes to the language which describe the limitations of their findings and clarify the methods. The work remains of interest to the field. I feel that they have sufficiently addressed my comments/concerns.

Specific comments

None

Authors: We thank both reviewers for their extensive and illuminating comments. We believe their recommendations and criticisms led to a better paper and strengthened the arguments we propose, along with fixing certain issues. Below, we responded to every reviewer comment point by point. We color coded our responses in red and the changes in the manuscript in blue. In addition, in the main manuscript we highlighted the changes in the paper.

Reviewers' comments:

Reviewer #1 (Remarks to the Author):

Review: Intrinsic Neural Timescales Flexibly Mediate Behavioral States

This work by Catal and colleagues investigates how intrinsic neural timescales (INT) mediate task states during resting flexibly. To address this, authors use calcium imaging data from rodents during rest and locomotion as well as human EEG recorded during cognitive tasks involving self-and-non-self-related processing. Finally, they use a neural mass model to investigate the underlying mechanisms to explain empirical observations during rest and task states with computational modelling. They have reported few primary findings: (i) Behavioral states during tasks in both rodent and human data can be accurately predicted from INT, (ii) INT becomes longer during task states compared to rest, (iii) INT changes from rest to task is correlated negatively with the variability of INT during rest, (iv) neural mass modelling shows a key role of recurrent connections in mediating the rest - task change of INT.

Although the questions raised here are sufficiently interesting empirically and computationally and may benefit diverse research areas that investigate the dynamic nature of the brain's INT in shaping behaviour and brain states, there are also many areas of real concern for this work. Although this cross-species attempt to address heterogeneity and variability of neural time scales is timely from the outset, and their correlation with behavior looks exciting and interesting, fundamentally speaking, there are some major drawbacks and loopholes that the authors do not address with scientific rigour to draw overarching conclusions what they would like to reach. Below, I outline my detailed comments, major problems, and suggestions to help authors review these points carefully while revising.

Major comments

1) Is INT modulating stimulus- or task-related activity, or could something else (underlying biological substrate) modulate intrinsic neural time scales derived from brain signals that may constrain stimulus-driven activity in a specific way? Please consider rephrasing this in the abstract.

Authors: We agree with this excellent comment. Indeed, INT recorded via EEG in human subjects and via calcium imaging in mice correspond to biological processes or substrates when considered on another epistemic level. Phrased differently, we can look at the same process on a smaller level of the biological substrate or on another level of dynamics, which, in turn, we measure via EEG and calcium imaging. Our study focuses on the level of timescales (INT) that we attempt to capture or measure via the signal's autocorrelation decay. You are correct that this leaves open what biological processes (on a smaller level) constitute the length of INT and

changes in INT between rest and task states. Therefore, the question of what biological processes underlie INT and how these biological processes mediate INT changes during task states is interesting but reaches beyond the capabilities of what we can measure with EEG and calcium imaging. In order to address this issue, we, in a first very tentative step, included the computational modelling suggesting that INT may be linked to recurrent processing.

We now address this point throughout the manuscript. In neuroimaging, it remains an open question what kind of underlying biological processes drive INT during rest and how and which biological processes mediate INT changes during task states.

Title: Flexibility of Intrinsic Neural Timescales During Distinct Behavioral States

Abstract: The sentence “At the same time, neural timescales also modulate stimulus- or task-related activity.”

was changed to ““At the same time, neural timescales also reflect stimulus- or task-related activity, showing changes specific to the task and behavior.”.

The sentence “Extending current findings, our results show the dynamic nature of the brain’s INT in shaping continuous naturalistic behavior through rest-task modulation possibly mediated by recurrent connections.”

Was also changed with “Extending current findings, our results show the dynamic nature of the brain’s INT in reflecting continuous naturalistic behavior through rest-task modulation possibly mediated by recurrent connections.”.

Introduction:

“Recent neuroimaging work has shown that INT modulate various aspects of cognitive processing. Examples of tasks for this modulation include but not limited to reward, self, narrative construction, consciousness and music listening.”

was changed to “Recent neuroimaging work has shown that INT exhibit changes during various aspects of cognitive processing. Examples of tasks showing INT changes include reward, self, narrative construction, consciousness and music listening.”.

The sentence “Together, our findings show that INT changes flexibly modulate behavioral states through rest task interaction as possibly mediated by recurrent connections in the brain.”

was changed by “Together, our findings show the flexible nature of the INT during continuous naturalistic behavioral states through their rest-task interaction possibly mediated by recurrent connections in the brain.”.

Conclusion:

The sentence “Biological organisms show highly flexible behavior. How this flexibility of behavior is mediated by the brain’s intrinsic neural timescales (INT) is an open question.”

is replaced with “Biological organisms show highly flexible behavior along different timescales. How this flexibility of continuous behavior relate to the brain’s intrinsic neural timescales (INT) is an open question.”

“Our results support and add to the previous findings on the behavioral relevance of INTs by showing their key role in the flexible rest task modulation of behavior possibly through recurrent connections.”

is replaced with “Our results support and add to the previous findings on the behavioral relevance of INTs by showing their dynamic changes during distinct continuous behavioral states as possibly mediated by recurrent connections.”

2) The authors cite recent neuroimaging work in the introduction to motivate the rationale for examining INT modulation during goal-directed behavior/task. However, I am very concerned and hesitant to accept that INT modulates various aspects of cognitive processing in myriad task scenarios. What is true is the heterogeneity & hierarchy of time scales and their association with spontaneous brain-wide activity and task-specific modulation of INT, as shown by previous primate and human studies. However, the converse is difficult to digest, stating that INT modulates goal-directed tasks or task switching. Even if we accept the example, the author cites a conversation composed of words, sentences, or perhaps higher-order stimuli such as music associated with different temporal receptive windows (TRWs) of processing specific brain regions of interest or involves distributed processing in the salient brain regions. The underlying mechanisms that modulate TRWs across sensory and cognitive processing remain elusive. The present introduction requires more clarity to situate the relevant research questions they are investigating, or the scope is too narrow. Listening to real-life conversations, auditory narratives, music, watching movies, and all naturalistic tasks involves complex processing. The only change in time-scale features alone predicts complex cognitive functions that are difficult to digest. I would rather think similarly to what was proposed by Gao et al., 2019 (the authors have cited) that INT is functionally very dynamic and constrained by underlying structural properties of neurons in the cortex, particularly during spontaneous activity, and perhaps, get more finely tuned during specific tasks.

Gao, R., van den Brink, R. L., Pfeffer, T. & Voytek, B. Neuronal timescales are functionally dynamic and shaped by cortical microarchitecture. *eLife* 9, e61277 (2020).

Also, very similar findings here in the context of neurodevelopment and atypical neural timescales by Watanabe et al., 2019

Watanabe, T., Rees, G., & Masuda, N. (2019). Atypical intrinsic neural timescale in autism. *Elife*, 8, e42256.

Authors: We agree with the excellent point that a general modulation of cognition and behavior by rest INT is likely an overgeneralization of our and other studies’ results. Hence, we agree that we need to soften the interpretation of our results to avoid the results’ overgeneralization. Paradigmatically and shown by several empirical neuroimaging studies, resting state INT can be related to cognition in various tasks (for example, Honey et al., 2012; Wolman et al., 2023; Ventura et al., 2024; Cavanagh et al., 2020; Zeraati et al., 2023). However, the directionality of the relationship, i.e. from cognitive task to INT or from INT to cognitive task cannot be asserted

based on our analysis of their statistical relations. We do very much agree that it is indeed very feasible to assume that the underlying structural properties of neurons in the cortex constrain the INT which, in turn, allow for their fine tuning during the specific tasks, as the reviewer suggests. To emphasize this discussion, we add the following paragraph to the paper. In addition, we make the changes mentioned below throughout the manuscript.

Discussion: Behavioral States Reflect INT Dynamics

Here, we should note that while we describe a relationship between INT and its specificity to behavior, we do not make any assertion about either causality let alone about the directionality of this relationship. As shown by several empirical neuroimaging studies, resting state INT can be related to cognition in various tasks (for example, Honey et al., 2012; Wolman et al., 2023; Ventura et al., 2024; Cavanagh et al., 2020; Zeraati et al., 2023). However, the causal nature of this relationship and its directionality, i.e. from cognitive task to INT or from INT to cognitive task cannot be asserted based solely on our analysis of their statistical relations. It is feasible to assume that the underlying structural properties of neurons in the cortex constrain the INT which, in turn, are more fine-tuned during specific tasks (Gao et al., 2020; Watanabe et al., 2019).

In order to clarify our intent, we change the following sentence in the introduction:

Previously: Building on these observations of rest – task modulation, we ask whether an analogous modulation of INT between rest and behavioral states holds.

Is changed to: Building on these observations of rest – task modulation of neural activity in its amplitude, variability and functional connectivity, we ask whether the INT modulate the transition from rest to continuous behavioral task states.

In addition to these changes, we believe that the changes made for the first comment should also clarify our stance.

References:

Honey, Christopher J., Thomas Thesen, Tobias H. Donner, Lauren J. Silbert, Chad E. Carlson, Orrin Devinsky, Werner K. Doyle, Nava Rubin, David J. Heeger, and Uri Hasson. "Slow Cortical Dynamics and the Accumulation of Information over Long Timescales." *Neuron* 76, no. 2 (October 18, 2012): 423–34. <https://doi.org/10.1016/j.neuron.2012.08.011>.

Wolman, Angelika, Yasir Çatal, Annemarie Wolff, Soren Wainio-Theberge, Andrea Scalabrini, Abdessadek El Ahmadi, and Georg Northoff. "Intrinsic Neural Timescales Mediate the Cognitive Bias of Self - Temporal Integration as Key Mechanism." *NeuroImage* 268 (March 2023): 119896. <https://doi.org/10.1016/j.neuroimage.2023.119896>.

Ventura, Bianca, Yasir Çatal, Angelika Wolman, Andrea Buccellato, Austin Clinton Cooper, and Georg Northoff. "Intrinsic Neural Timescales Exhibit Different Lengths in Distinct Meditation Techniques." *NeuroImage* 297 (August 15, 2024): 120745. <https://doi.org/10.1016/j.neuroimage.2024.120745>.

Cavanagh, Sean E., Laurence T. Hunt, and Steven W. Kennerley. "A Diversity of Intrinsic Timescales Underlie Neural Computations." *Frontiers in Neural Circuits* 14 (2020). <https://www.frontiersin.org/articles/10.3389/fncir.2020.615626>.

Zeraati, Roxana, Yan-Liang Shi, Nicholas A. Steinmetz, Marc A. Gieselmann, Alexander Thiele, Tirin Moore, Anna Levina, and Tatiana A. Engel. "Intrinsic Timescales in the Visual Cortex Change with Selective Attention and Reflect Spatial Connectivity." *Nature Communications* 14, no. 1 (April 3, 2023): 1858. <https://doi.org/10.1038/s41467-023-37613-7>.

Gao, Richard, Ruud L van den Brink, Thomas Pfeffer, and Bradley Voytek. "Neuronal Timescales Are Functionally Dynamic and Shaped by Cortical Microarchitecture." Edited by Martin Vinck, Laura L Colgin, and Thilo Womelsdorf. *eLife* 9 (November 23, 2020): e61277. <https://doi.org/10.7554/eLife.61277>.

Watanabe, Takamitsu, Geraint Rees, and Naoki Masuda. "Atypical Intrinsic Neural Timescale in Autism." Edited by Joshua I Gold, Michael Breakspear, and Leonardo L Gollo. *eLife* 8 (February 5, 2019): e42256. <https://doi.org/10.7554/eLife.42256>.

3) A related point to the above is debatable; as the authors mentioned, the current work indicates that INT becomes longer with task engagement and increased attention during behavior, as shown recently by Zeraati, R. et al.2023 (cited by the authors) also suggests that increased attention and stimulus engagement, which modulates INT not the reverse that INT causally increases attention and stimulus engagement reinforcing what was mentioned earlier.

Authors: This is an excellent point. We agree that Zeraati et al. (2023) demonstrated that attention can correlate or overlap with changes in INT. We do not mean to imply that INT causally lead to changes in cognition or behavior. Neither do we mean to imply that cognition or behavior causally led to changes in INT. Instead, we deal with two epistemic levels here: one level is neuronal dynamics, and the other level is psychological states, such as attention. What our study shows is that changes in rest-to-task INT are related to changes in behavior. Accordingly, we do not imply causal relation from INT (as perceived from a level of neuronal dynamics) to behavior (as perceived from the level of psychology). We thus agree with you that our writing did not make this important distinction clear and therefore likely leads to confusion for the readers of our paper. We thus softened our hypotheses and discussion of the results, where we accordingly talk about relationship (or correspondence) and explicitly refrain from any implications of a causal chain from INT to behavior.

We believe that the paragraph we added to the discussion for the point 2 above also answers this comment. For completeness, we copy it below.

Discussion: Behavioral States Reflect INT Dynamics

Here, we should note that while we describe a relationship between INT and its specificity to behavior, we do not make any assertion about either causality let alone about the directionality of this relationship. As shown by several empirical neuroimaging studies, resting state INT can be related to cognition in various tasks (for example, Honey et al., 2012; Wolman et al., 2023; Ventura et al., 2024; Cavanagh et al., 2020; Zeraati et al., 2023). However, the causal nature of this relationship and its directionality, i.e. from cognitive task to INT or from INT to cognitive task cannot be asserted based solely on our analysis of their statistical relations. It is feasible to assume that the underlying structural properties of neurons in the cortex constrain the INT which, in turn, are more fine-tuned during specific tasks (Gao et al., 2020; Watanabe et al., 2019).

Furthermore, we add the following in the same discussion section that further explicates Zeraati's findings:

Previously: The active tracking requires integration of information over longer timescales, which reflects our findings of longer timescales during the narrative listening compared to rest. Such prolongation of the timescales is in agreement with studies showing that tasks that require heightened attentional focus elicit INT increase (Gao et al., 2020; Zeraati et al., 2023).

Is changed to: The active tracking requires integration of information over longer timescales, which reflects our findings of longer timescales during the narrative listening compared to rest. Such prolongation of the timescales converges with studies showing that tasks requiring heightened attentional focus elicit INT increase (Gao et al., 2020; Zeraati et al., 2023). Zeraati et al. explained the increase of INT by a neurophysiological mechanism involving acetylcholine, which increases vertical neural interactions while decreasing horizontal neural interactions. This serves as an example of how INT is altered both by external input, e.g., a task with stimuli, and by neurophysiological changes.

References

Honey, Christopher J., Thomas Thesen, Tobias H. Donner, Lauren J. Silbert, Chad E. Carlson, Orrin Devinsky, Werner K. Doyle, Nava Rubin, David J. Heeger, and Uri Hasson. "Slow Cortical Dynamics and the Accumulation of Information over Long Timescales." *Neuron* 76, no. 2 (October 18, 2012): 423–34. <https://doi.org/10.1016/j.neuron.2012.08.011>.

Wolman, Angelika, Yasir Çatal, Annemarie Wolff, Soren Wainio-Theberge, Andrea Scalabrini, Abdessadek El Ahmadi, and Georg Northoff. "Intrinsic Neural Timescales Mediate the Cognitive Bias of Self - Temporal Integration as Key Mechanism." *NeuroImage* 268 (March 2023): 119896. <https://doi.org/10.1016/j.neuroimage.2023.119896>.

Ventura, Bianca, Yasir Çatal, Angelika Wolman, Andrea Buccellato, Austin Clinton Cooper, and Georg Northoff. "Intrinsic Neural Timescales Exhibit Different Lengths in Distinct Meditation Techniques." *NeuroImage* 297 (August 15, 2024): 120745. <https://doi.org/10.1016/j.neuroimage.2024.120745>.

Cavanagh, Sean E., Laurence T. Hunt, and Steven W. Kennerley. "A Diversity of Intrinsic Timescales Underlie Neural Computations." *Frontiers in Neural Circuits* 14 (2020). <https://www.frontiersin.org/articles/10.3389/fncir.2020.615626>.

Zeraati, Roxana, Yan-Liang Shi, Nicholas A. Steinmetz, Marc A. Gieselmann, Alexander Thiele, Tirin Moore, Anna Levina, and Tatiana A. Engel. "Intrinsic Timescales in the Visual Cortex Change with Selective Attention and Reflect Spatial Connectivity." *Nature Communications* 14, no. 1 (April 3, 2023): 1858. <https://doi.org/10.1038/s41467-023-37613-7>.

Gao, Richard, Ruud L van den Brink, Thomas Pfeffer, and Bradley Voytek. "Neuronal Timescales Are Functionally Dynamic and Shaped by Cortical Microarchitecture." Edited by Martin Vinck, Laura L Colgin, and Thilo Womelsdorf. *eLife* 9 (November 23, 2020): e61277. <https://doi.org/10.7554/eLife.61277>.

Watanabe, Takamitsu, Geraint Rees, and Naoki Masuda. "Atypical Intrinsic Neural Timescale in Autism." Edited by Joshua I Gold, Michael Breakspear, and Leonardo L Gollo. *eLife* 8 (February 5, 2019): e42256. <https://doi.org/10.7554/eLife.42256>.

4) Author states, "How the INT in naturalistic behavioral states relates to the timescales of the brain's spontaneous activity remains yet unclear. Addressing this question is the goal of this study. However, I would like to ask what is ambiguous: what do they mean by behavioral states, and what is the difference between behavioral and naturalistic states? What are the key differences between behavioral states and brain states? There have been numerous occasions when these terms have been introduced in neuroscience to interpret different things and are very debatable, hence, an operational definition/meaning is much needed to offer any insight.

Authors: We agree that indeed the term "naturalistic" in neuroimaging paradigms can be problematic since it often lacks a precise definition, as discussed in Sonkusare et al. (2019). We agree that we also lack better definition. Instead of coming up with yet another attempt that tries to define and defend why our behavioral states are naturalistic, we would rather like to drop the term. Instead, we decided to talk about continuous behavior or behavioral states (Huk et al. 2018), as we believe it is highly debatable to what extent our behavioral states in the human and mouse datasets are naturalistic. Therefore, we removed the term 'naturalistic' throughout the paper. Regarding the definition of behavioral states: we suggest the following:

A behavioral state refers to a distinct pattern of behavior that an organism exhibits in response to internal and external stimuli (Flavell et al., 2020). In our case, the behavioral states are various modes of behavior of mice (such as initial rest, locomotion etc.) and the three behavioral states in humans.

Regarding the definition of a brain state: we admit that an agreed-on definition of the brain state is currently lacking (Kringelbach and Deco, 2020); nonetheless, we can assume the following more pragmatic definition: a brain state is characterized by a widely distributed pattern of neural activity (Greene et al., 2023). Operationally, we assume that such widely distributed pattern of activity consists in our case in the INT topography across the brain – this describes a spatial pattern of a brain state. Further, INT is operationalized by the decay rate of the autocorrelation function – this implies a temporal a pattern of activity of a brain state.

Therefore, the difference between the behavioral state and the brain state is that the behavioral state refers to which (behavioral) pattern of activity the organism exhibits in relation to its respective environmental context, i.e., what it is doing, whereas the notion of a brain state refers to which neural pattern of activity the organism's brain is in, i.e. what is happening in the brain.

To clarify these points, we added the following to the introduction, after the paragraph:

First and foremost, we may want to do some conceptual homework. What do we mean by the notions of behavioral state and brain state? A behavioral state refers to a distinct pattern of locomotor activity, cognition, and physiological activity that an organism exhibits in response to internal and external stimuli within a particular environmental context (Flavell et al., 2020). An agreed upon definition of a brain state is currently lacking (Kringelbach and Deco, 2020), nonetheless, we assume the following definition: a brain state is characterized by a widely distributed pattern of activity (Greene et al., 2023) which can be specified by its spatial pattern (like its topographic distribution) and temporal pattern (like its changes or non-changes over time).

References

Sonkusare, Saurabh, Michael Breakspear, and Christine Guo. "Naturalistic Stimuli in Neuroscience: Critically Acclaimed." *Trends in Cognitive Sciences* 23, no. 8 (August 2019): 699–714. <https://doi.org/10.1016/j.tics.2019.05.004>.

Huk, Alexander, Kathryn Bonnen, and Biyu J. He. "Beyond Trial-Based Paradigms: Continuous Behavior, Ongoing Neural Activity, and Natural Stimuli." *The Journal of Neuroscience: The Official Journal of the Society for Neuroscience* 38, no. 35 (August 29, 2018): 7551–58. <https://doi.org/10.1523/JNEUROSCI.1920-17.2018>.

Flavell, Steven W., Nadine Gogolla, Matthew Lovett-Barron, and Moriel Zelikowsky. "The Emergence and Influence of Internal States." *Neuron* 110, no. 16 (August 17, 2022): 2545–70. <https://doi.org/10.1016/j.neuron.2022.04.030>.

Kringelbach, Morten L., and Gustavo Deco. "Brain States and Transitions: Insights from Computational Neuroscience." *Cell Reports* 32, no. 10 (September 8, 2020): 108128. <https://doi.org/10.1016/j.celrep.2020.108128>.

Greene, Abigail S., Corey Horien, Daniel Barson, Dustin Scheinost, and R. Todd Constable. "Why Is Everyone Talking about Brain State?" *Trends in Neurosciences* 46, no. 7 (July 2023): 508–24. <https://doi.org/10.1016/j.tins.2023.04.001>.

5) The author cites a handful of selective studies to indicate that modulation of task INT by rest INT is motivated by a wide range of studies demonstrating that the dynamics of rest or pre-stimulus state can modulate the brain's response to an external stimulus. On the contrary, I can say numerous studies show the brain's intrinsic dynamics during resting state are very different from different categories of sensory and stimulus-driven goal-directed tasks, and often, prestimulus brain states have no bearing on predicting stimulus-induced response/behavior. Consequently, giving null results. Also, the dimensionality of resting-state brain states differs greatly from task-evoked states operating on a low-dimensional subspace. Hence, I would, at best, say the evidence is mixed, and it is too hasty to postulate a connection/associate to put forward a hypothesis based on selective sampling of a few studies borrowed from primate electrophysiology, fMRI and EEG vastly operating on different physiological time scales.

Authors: We agree with the reviewer that our view is not the only view, but that an alternative theory where resting dynamics is orthogonal to and thus unrelated with task dynamics is also possible. Hence, we agree that this an overgeneralization on our side which we now corrected accordingly. Below, we added new references supporting both sides of the argument and color – coded them with purple.

Previously: Our focus on the modulation of task INT by rest INT is motivated by a wide range of studies demonstrating that the dynamics of rest or pre-stimulus state can modulate the brain's response to an external stimulus. Various studies in functional magnetic resonance imaging (fMRI) (Fox et al., 2005; Boly et al., 2007; Hesselman et al., 2008; Mennes et al., 2010; Coste et al., 2011; He et al., 2013; Sadaghiani et al., 2013 and 2015; Huang et al., 2017; Çatal et al., 2022), electroencephalography (EEG) (Wolman et al., 2023; Wolff et al., 2019 and 2021; Kolvoort et al., 2020; Wainio – Theberge et al., 2021) and single-unit neural recordings (Petersen et al., 2003; Churchland et al., 2010; Braun et al., 2022) show an active modulation of task-related activity by the brain's spontaneous activity (see also Northoff et al., 2010a and 2010b for a review).

Is changed to: Our focus on the modulation of task INT by rest INT is motivated by a wide range of studies demonstrating that the dynamics of rest or pre-stimulus state can modulate the brain's response to an external stimulus. Various studies in functional magnetic resonance imaging (fMRI) (Fox et al., 2005; Boly et al., 2007; Hesselmann et al., 2008; Mennes et al., 2010; Coste et al., 2011; He et al., 2013; Sadaghiani et al., 2009, 2013 and 2015; Huang et al., 2017; Çatal et al., 2022; Kamp et al., 2018; Wu et al., 2024), electroencephalography (EEG) (Wolman et al., 2023; Wolff et al., 2019 and 2021; Kolvoort et al., 2020; Wainio – Theberge et al., 2021; Waschke et al., 2017; Iemi et al., 2017 and 2022; Romei et al., 2008) and single-unit neural recordings (Petersen et al., 2003; Churchland et al., 2010; Braun et al., 2022) show an active modulation of task-related activity by the brain's spontaneous activity (see also Northoff et al., 2010a and 2010b for a review). Although it should be mentioned that other findings noted no relationship or the divergence between spontaneous and task – evoked activity (Avitan et al., 2021, 2022; Cowley et al., 2016; Stringer et al., 2019; Triplett et al., 2020): these studies show that the dimensionality of the resting state dynamics differs greatly from task evoked states with little or no correlation of rest and task states – in that case, task-related and spontaneous activity are orthogonal to each other. Whether this difference is physiological, as related to distinct kinds of tasks, or methodological, as related to different analyses methods (Wolff et al. 2019, 2021) remains yet to be clarified and tested. We here follow the analyses methods of the first stance, the relationship of spontaneous and task activity and develop our hypotheses accordingly.

References

- Fox, M. D. et al. The human brain is intrinsically organized into dynamic, anticorrelated functional networks. *Proceedings of the National Academy of Sciences* 102, 9673–9678 (2005).
- Boly, M. et al. Baseline brain activity fluctuations predict somatosensory perception in humans. *Proceedings of the National Academy of Sciences* 104, 12187–12192 (2007).
- Hesselmann, G., Kell, C. A., Eger, E. & Kleinschmidt, A. Spontaneous local variations in ongoing neural activity bias perceptual decisions. *Proc Natl Acad Sci U S A* 105, 10984–10989 (2008).
- Mennes, M. et al. Inter-individual differences in resting-state functional connectivity predict task-induced BOLD activity. *NeuroImage* 50, 1690–1701 (2010).
- Coste, C. P., Sadaghiani, S., Friston, K. J. & Kleinschmidt, A. Ongoing Brain Activity Fluctuations Directly Account for Intertrial and Indirectly for Intersubject Variability in Stroop Task Performance. *Cerebral Cortex* 21, 2612–2619 (2011).
- He, B. J. Spontaneous and Task-Evoked Brain Activity Negatively Interact. *J. Neurosci.* 33, 4672–4682 (2013).
- Sadaghiani, S. & Kleinschmidt, A. Functional interactions between intrinsic brain activity and behavior. *NeuroImage* 80, 379–386 (2013).
- Sadaghiani, S., Poline, J.-B., Kleinschmidt, A. & D'Esposito, M. Ongoing dynamics in large-scale functional connectivity predict perception. *Proceedings of the National Academy of Sciences* 112, 8463–8468 (2015).

Huang, Z. et al. Is There a Nonadditive Interaction Between Spontaneous and Evoked Activity? Phase-Dependence and Its Relation to the Temporal Structure of Scale-Free Brain Activity. *Cerebral Cortex* 27, 1037–1059 (2017).

Çatal, Y., Gomez-Pilar, J. & Northoff, G. Intrinsic dynamics and topography of sensory input systems. *Cerebral Cortex* 32, 4592–4604 (2022).

Wolff, A. et al. Prestimulus dynamics blend with the stimulus in neural variability quenching. *NeuroImage* 238, 118160 (2021).

Wolff, A. et al. Neural variability quenching during decision-making: Neural individuality and its prestimulus complexity. *NeuroImage* 192, 1–14 (2019).

Kolvoort, I. R., Wainio-Theberge, S., Wolff, A. & Northoff, G. Temporal integration as “common currency” of brain and self-scale-free activity in resting-state EEG correlates with temporal delay effects on self-relatedness. *Human Brain Mapping* 41, 4355–4374 (2020).

Wainio-Theberge, S., Wolff, A. & Northoff, G. Dynamic relationships between spontaneous and evoked electrophysiological activity. *Commun Biol* 4, 1–17 (2021).

Petersen, C. C. H., Hahn, T. T. G., Mehta, M., Grinvald, A. & Sakmann, B. Interaction of sensory responses with spontaneous depolarization in layer 2/3 barrel cortex. *Proceedings of the National Academy of Sciences* 100, 13638–13643 (2003).

Churchland, M. M. et al. Stimulus onset quenches neural variability: a widespread cortical phenomenon. *Nat Neurosci* 13, 369–378 (2010).

Braun, W., Matsuzaka, Y., Mushiake, H., Northoff, G. & Longtin, A. Non-additive activity modulation during a decision making task involving tactic selection. *Cogn Neurodyn* 16, 117–133 (2022).

Northoff, G., Qin, P. & Nakao, T. Rest-stimulus interaction in the brain: a review. *Trends in Neurosciences* 33, 277–284 (2010).

Northoff, G., Duncan, N. W. & Hayes, D. J. The brain and its resting state activity—Experimental and methodological implications. *Progress in Neurobiology* 92, 593–600 (2010).

Sadaghiani, Sepideh, Guido Hesselmann, and Andreas Kleinschmidt. “Distributed and Antagonistic Contributions of Ongoing Activity Fluctuations to Auditory Stimulus Detection.” *Journal of Neuroscience* 29, no. 42 (October 21, 2009): 13410–17.
<https://doi.org/10.1523/JNEUROSCI.2592-09.2009>.

Kamp, Tabea, Bettina Sorger, Caroline Benjamins, Lars Hausfeld, and Rainer Goebel. “The Prestimulus Default Mode Network State Predicts Cognitive Task Performance Levels on a Mental Rotation Task.” *Brain and Behavior* 8, no. 8 (June 22, 2018): e01034.
<https://doi.org/10.1002/brb3.1034>.

Wu, Yuan-Hao, Ella Podvalny, Max Levinson, and Biyu J. He. “Network Mechanisms of Ongoing Brain Activity’s Influence on Conscious Visual Perception.” *Nature Communications* 15, no. 1 (July 8, 2024): 5720. <https://doi.org/10.1038/s41467-024-50102-9>.

Waschke, Leonhard, Malte Wöstmann, and Jonas Obleser. "States and Traits of Neural Irregularity in the Age-Varying Human Brain." *Scientific Reports* 7, no. 1 (December 12, 2017): 17381. <https://doi.org/10.1038/s41598-017-17766-4>.

Iemi, Luca, Maximilien Chaumon, Sébastien M. Crouzet, and Niko A. Busch. "Spontaneous Neural Oscillations Bias Perception by Modulating Baseline Excitability." *The Journal of Neuroscience: The Official Journal of the Society for Neuroscience* 37, no. 4 (January 25, 2017): 807–19. <https://doi.org/10.1523/JNEUROSCI.1432-16.2016>.

Iemi, Luca, Laura Gwilliams, Jason Samaha, Ryszard Aukstulewicz, Yael M Cycowicz, Jean-Remi King, Vadim V Nikulin, et al. "Ongoing Neural Oscillations Influence Behavior and Sensory Representations by Suppressing Neuronal Excitability." *NeuroImage* 247 (February 15, 2022): 118746. <https://doi.org/10.1016/j.neuroimage.2021.118746>.

Romei, Vincenzo, Verena Brodbeck, Christoph Michel, Amir Amedi, Alvaro Pascual-Leone, and Gregor Thut. "Spontaneous Fluctuations in Posterior Alpha-Band EEG Activity Reflect Variability in Excitability of Human Visual Areas." *Cerebral Cortex (New York, N.Y.: 1991)* 18, no. 9 (September 2008): 2010–18. <https://doi.org/10.1093/cercor/bhm229>.

Avitan, Lilach, Zac Pujic, Jan Mölter, Shuyu Zhu, Biao Sun, and Geoffrey J Goodhill. "Spontaneous and Evoked Activity Patterns Diverge over Development." Edited by Tatyana O Sharpee, Timothy E Behrens, and Emre Yaksi. *eLife* 10 (April 19, 2021): e61942. <https://doi.org/10.7554/eLife.61942>.

Avitan, Lilach, and Carsen Stringer. "Not so Spontaneous: Multi-Dimensional Representations of Behaviors and Context in Sensory Areas." *Neuron* 110, no. 19 (October 5, 2022): 3064–75. <https://doi.org/10.1016/j.neuron.2022.06.019>.

Triplett, Marcus A., Zac Pujic, Biao Sun, Lilach Avitan, and Geoffrey J. Goodhill. "Model-Based Decoupling of Evoked and Spontaneous Neural Activity in Calcium Imaging Data." *PLOS Computational Biology* 16, no. 11 (November 30, 2020): e1008330. <https://doi.org/10.1371/journal.pcbi.1008330>.

Cowley, Benjamin R., Matthew A. Smith, Adam Kohn, and Byron M. Yu. "Stimulus-Driven Population Activity Patterns in Macaque Primary Visual Cortex." *PLOS Computational Biology* 12, no. 12 (December 9, 2016): e1005185. <https://doi.org/10.1371/journal.pcbi.1005185>.

Stringer, Carsen, Marius Pachitariu, Nicholas Steinmetz, Matteo Carandini, and Kenneth D. Harris. "High-Dimensional Geometry of Population Responses in Visual Cortex." *Nature* 571, no. 7765 (July 2019): 361–65. <https://doi.org/10.1038/s41586-019-1346-5>.

6) If I understood the authors correctly, then they are simply trying to answer the question of whether INT is behaviourally relevant or not. In that case, the pertinent question comes how a task that is carried out by mice in spontaneous awake condition, such as their calcium imaging data recorded during spontaneous behavior, relates to human EEG data recorded from participants while they listen to autobiographical (self-related) and non-autobiographical (non-self / other related) narratives. I am not fully convinced they are related.

Authors: This is an excellent description of our intention. Yes, of course, the two task states are different. The two datasets' task states' nonetheless share one fundamental factor on a deeper

level that is essential to INT, precisely that they offer continuous (non-interrupted) measurement of the continuous behavior featuring both tasks (Huk et al., 2018): INT attempt to measure timescales, hence they rest on a continuous process that underlies both the listening of narratives in the case of humans and the motor output by the mice. The two tasks do indeed come with different psychological requirements (moving vs listening). However, on a deeper more physiological level, they both share the need for the continuity of their underlying neural activity which can be measured by the continuity as operationalized by the INT. We improved our paper by highlighting and better explaining this issue. Furthermore, our usage of the term behavior here is consistent with the literature (see for example Kay et al., 2023: figure 1 and its caption; Barch et al. (2014): tasks such as audition, working memory, verbal episodic memory are classified under behavioral task measures in tables 2 and 3; even though collected outside the scanner). Therefore, we suggest that both tasks converge under the definition of a continuous behavioral state.

To clarify the issue, we added the following paragraphs to introduction and discussion:

Introduction:

On the more methodological side, we should note that while the psychological requirements of the two tasks (motor outputs versus self – evaluation) differ from each other, they nevertheless share a fundamental factor that is essential to INT: both tasks require continuous, that is, non – interrupted behavior which distinguishes the task states in both mice and humans from their respective resting state that does not require such continuous non-interrupted activity – this is measured by quantifying the temporal continuity of their underlying neural activity by the INT.

Discussion:

Behavioral states reflect INT dynamics

“Before going further, we should make clear the similar and different aspects of the tasks for mice and humans. The exact task details and their psychological requirements are rather distinct in mice and humans: a spontaneous behavioral motion task for mice and a self – evaluation task for humans. On a deeper level, there is a similarity consisting of that both tasks are continuously ongoing and thus non-interrupted during which time course they yield distinct behavioral states (like the distinct behavioral periods of the mice and the distinct varying ratings in humans). We are aware that our use of the word ‘behavioral’ can be criticized here if one constrains the definition of behavior to the behavior of whole body, as in mice which would exclude to conceive the cursor movement in the human task as a behavioral state. However, classifying the cursor movement as a behavior is consistent with the usage of the term behavior in the literature (see for example figure 1 of Kay et al. (2023) and tables 2 and 3 of Barch et al. (2013)). Together, we suggest that the temporal continuity in the behavior of both tasks can be captured by the temporal continuity of their underlying neuronal activity which can be measured by the INT as they are based on the autocorrelation function.”

References

Huk, Alexander, Kathryn Bonnen, and Biyu J. He. “Beyond Trial-Based Paradigms: Continuous Behavior, Ongoing Neural Activity, and Natural Stimuli.” *The Journal of Neuroscience: The Official Journal of the Society for Neuroscience* 38, no. 35 (August 29, 2018): 7551–58. <https://doi.org/10.1523/JNEUROSCI.1920-17.2018>.

Kay, Kendrick, Kathryn Bonnen, Rachel N. Denison, Mike J. Arcaro, and David L. Barack. "Tasks and Their Role in Visual Neuroscience." *Neuron* 111, no. 11 (June 7, 2023): 1697–1713. <https://doi.org/10.1016/j.neuron.2023.03.022>.

Barch, Deanna M., Gregory C. Burgess, Michael P. Harms, Steven E. Petersen, Bradley L. Schlaggar, Maurizio Corbetta, Matthew F. Glasser, et al. "Function in the Human Connectome: Task-fMRI and Individual Differences in Behavior." *NeuroImage* 80 (May 16, 2013): 169. <https://doi.org/10.1016/j.neuroimage.2013.05.033>.

7) Secondly, any number of training features can be used to train a classifier model (they use a rather simplistic ML model like SVM). This is fine. However, the only handcrafted feature they use here is INT. Hence, method-wise, I do not see the rationale for using a classifier instead of simply using multivariate regression models based on INTs estimated empirically from brain areas (ROIs) to predict behavioral response (dependent variables). What the authors have done instead is a classification task to see whether the true labels versus labels predicted by the classifier model agree to what degree of accuracy. Suppose they already know the class labels, e.g., onset of locomotion, locomotion, the offset of locomotion, initial rest, and sustained rest. Wouldn't it be better to employ a prediction approach according to their whole-brain INT estimates rather than a classification task to classify behavioral states? I would suggest that the authors review their approach.

Authors: We thank the reviewer for this excellent comment. The criticism of no explicit rationale for SVM is a fair one. We added logistic regressions to the paper to close this gap. We hope the following changes and additions to methods, results and supplementary material, should resolve this challenge:

Methods

While SVM is a powerful tool for classification, the nonlinear transformation it performs on the data using a radial basis kernel makes the results more difficult to interpret. In addition to SVM, we performed logistic regression as well. Logistic regression models the probabilities of each behavioral state as a linear combination of predictors, therefore it is easier to interpret. The data is organized as the same way as SVM. As in SVM, we used nested cross – validation with 10 inner and 10 outer folds to tune the hyperparameter lambda, denoting regularization strength. We set the relative coefficient tolerance for terminating the optimization to 0.0001. We report confusion matrix and one – versus – all ROC curve calculated on test data and aggregate results from 10 outer folds for confusion matrix, this is the same as in SVM. The averaged ROC curves and aggregate confusion matrix can be found in supplementary figures 7, 24. The ROC curves and confusion matrices for each fold can be found in supplementary figures 8, 9, 25, 26.

Results

Behavioral states in mice are distinguished by their intrinsic neural timescales

In addition to SVM, we also performed logistic regression as a prediction task as opposed to classification of SVM. We fitted the logistic regression model in the same way as in SVM: using 10-fold nested cross validation. Supplementary figures 7, 8 and 9 show the average ROC curve and aggregate confusion matrix, that is, confusion matrices for individual folds and ROC curves

for individual folds respectively. We evaluated the model's performance on test data as in SVM. The regression model showed an overall accuracy of 86%. For every state, the ROC curves have an AUC above 0.9.

Replication of the Mice Results in a Human EEG Dataset of Self-Evaluation

We additionally used logistic regression in addition to SVM. Supplementary figures 24, 25 and 26 show the results for average ROC curve and aggregate confusion matrix, confusion matrices for individual folds and ROC curves for individual folds respectively. The overall accuracy of the model evaluated in the test set is 58%. The ROC curves have an AUC value above 0.7 for all states.

Supplementary Material - Logistic Regression Results for Mice Data

In addition to SVM, we performed logistic regression for prediction of behavioral states from data. Supplementary figure 7 shows the average ROC curve and aggregate confusion matrix evaluated on the test set. Supplementary figures 8 and 9 show individual ROC curves and confusion matrices for each fold.

Supplementary Figure 7. Classification learning for behavioral states using logistic regression instead of SVM. a. We trained logistic regression models to classify the behavioral states of mice based on the ACW-0 values across the brain with nested cross validation for hyperparameter tuning (10 inner, 10 outer folds). Panel a shows the confusion matrix for test data, aggregated across folds. b. Receiver operating characteristic curve for the test data using the logistic regression model. The data ROC curves from 10 folds were averaged and the standard deviation across the folds were indicated as shading. Abbreviations: TPR: True positive rate, FNR: False negative rate, AUC: Area under the curve, M: Mean, STD: Standard deviation

Supplementary Figure 8. Confusion matrices for each of the 10 outer folds (see above text and methods) for the mice calcium imaging data using logistic regression instead of SVM.

Supplementary Figure 9. ROC curves for each of the 10 outer folds (see above text and methods) for the mice calcium imaging data using logistic regression instead of SVM. AUC: Area under the curve.

Supplementary Material – Logistic Regression Results for EEG Data

As explained above for the case of mice, we additionally did logistic regression for EEG data as well. Supplementary figure 24 shows the average ROC and aggregate confusion matrix. Supplementary figures 25 and 26 show confusion matrices and ROC curves for each fold.

Supplementary Figure 24. Classification learning for behavioral states using logistic regression instead of SVM in human EEG data. a. We trained logistic regression models to classify the behavioral states of mice based on the ACW-0 values across the brain with nested cross validation for hyperparameter tuning (10 inner, 10 outer folds). Panel a shows the confusion matrix for test data, aggregated across folds. b. Receiver operating characteristic curve for the test data using the logistic regression model. The data ROC curves from 10 folds were averaged and the standard deviation across the folds were indicated as shading. Abbreviations: TPR: True positive rate, FNR: False negative rate, AUC: Area under the curve, M: Mean, STD: Standard deviation

Supplementary Figure 25. Confusion matrices for each of the 10 outer folds (see above text and methods) for the human EEG data using logistic regression instead of SVM.

Supplementary Figure 26. ROC curves for each of the 10 outer folds (see above text and methods) for the human EEG data using logistic regression instead of SVM. AUC: Area under the curve.

8) Any possible confounds that may arise due to the nature of the self-versus non-self-task in human participants and their implications need to be mentioned in the methods and discussions. While taking cursor-based tracking of responses, particularly valence, how the participants have been instructed during the behavioral task is fundamentally important. What ensures the arousal experience in participants experienced due to the nature of the stimuli also needs to be verified by tracking behavioral parameters associated with this. Finally, I am very concerned about how the task designed for humans has anything to do with the rodent task determining behavioral and brain states in mice. At least, the authors have yet to show any convincing data to show at some level, neural and behavioral responses by humans and rodents are congruent. They seem to be very independent behavior measures in two different species. In my opinion, what binds this cross-species investigation is still somewhat unclear.

Authors: This comment is an important one. We address it in chunks, with the comments of the reviewer in black italic and ours in red.

Any possible confounds that may arise due to the nature of the self-versus non-self-task in human participants and their implications need to be mentioned in the methods and discussions.

One possible confounder we can think of is head motion, which is a challenge in practically every EEG study. Conventional methods to eliminate it are through preprocessing, which we applied thoroughly using algorithms that are widely accepted in the field (Winkler et al., 2014; which was used by Lopez et al., 2023; Gabard – Durnam et al., 2019; Zhang et al., 2023; Stephens and Zabelina, 2020; Prochnow et al., 2024 among many others). Nonetheless, it must be mentioned in the text; we added the following sentences in the Methods and Limitations parts of the paper:

Methods

Data Acquisition and Preprocessing

It needs to be mentioned that while we tried our best to eliminate any noise unrelated to brain activity during preprocessing via artifact rejection using standard and well established methods (Winkler et al., 2014; which was used by Lopez et al., 2023; Gabard – Durnam et al., 2019; Zhang et al., 2023; Stephens and Zabelina, 2020; Prochnow et al., 2024 among many others), we nevertheless cannot completely exclude motion artifacts as a possible confounder.

Limitations

Another limitation are possible confounding factors in the EEG task. The continuous movement of the mouse cursor might induce excessive motion artifacts. A higher head motion during any one of the tasks (self or non – self versus resting state) can influence INTs and alter the results. While we used standard well established methods for artifact rejection (Winkler et al., 2014; Lopez et al., 2023; Gabard – Durnam et al., 2019; Zhang et al., 2023; Stephens and Zabelina, 2020; Prochnow et al., 2024) to eliminate non – physiological noise from the data via artifact rejection, we cannot fully exclude that motion artifacts may still confound our EEG results.

While taking cursor-based tracking of responses, particularly valence, how the participants have been instructed during the behavioral task is fundamentally important.

We agree, and add the following sentences in the methods part:

Methods

Data Acquisition and Preprocessing

Old: “While listening, they indicated their internal states by moving the cursor (positive state towards the center, negative state: away from center.)”

New: “The subjects were instructed to continuously evaluate the narrative’s contents as positive or negative using a mouse cursor. Keeping the cursor close to the center indicated a negative content whereas away from center indicated positive, thus enabling them to track their internal state based on the narrative. The audios were presented with headphones. For ease of distinction, the cursor was colored green in the proximity of the center, red for away from center and yellow for intermediate distances. Additionally, all participants verbally confirmed their understanding of the cursor task instructions before beginning the experiment. “

What ensures the arousal experience in participants experienced due to the nature of the stimuli also needs to be verified by tracking behavioral parameters associated with this.

Unfortunately, we did not collect any additional behavioral parameters for arousal such as pupil size or skin conductance. However, we do not make any claims about arousal in the human EEG task in the paper. The basic structure of our analysis is about how the INT change during the behavioral states, and how this change relates to the resting INT dynamics. We agree with the reviewer that incorporation of behavioral parameters related to arousal would considerably strengthen the paper. In fact, this can be a future research direction. We added this issue to the limitations section of the paper and take note.

Limitations

Yet another limitation of the paper is a lack of behavioral measures reflecting arousal in human data. The addition of arousal would strengthen the arguments about behavioral relevance. The

relationship between arousal and INT in task states can be a future research avenue, to be explored further.

Finally, I am very concerned about how the task designed for humans has anything to do with the rodent task determining behavioral and brain states in mice. At least, the authors have yet to show any convincing data to show at some level, neural and behavioral responses by humans and rodents are congruent. They seem to be very independent behavior measures in two different species. In my opinion, what binds this cross-species investigation is still somewhat unclear.

We understand the concerns of the reviewer regarding the distinctness of behavioral states. Essentially, our argument boils down to the notion that both tasks are continuous tasks that elicit distinct behavioral states. While they imply different psychological requirements (self-evaluation versus spontaneous behavior), both tasks share some similarity on a deeper level, that is, in the temporal structure in that both require continuous non-interrupted activity, that is, temporal continuity – this is operationalized by measuring the temporal continuity of their underlying neural activity by the INT. Our results support such similarity on a deeper level: (i) the distinguishability of behavioral states in both mice and human tasks based on their INT topography, as shown by machine learning results, (ii) increase of INT in the behavioral states of both mice and humans compared to rest, (iii) negative correlation between rest variability of INT and the observed rest – task change in both mice and humans. Together, these converging findings supports our assertion that despite being psychologically distinct, both tasks share some similarity on a deeper most likely temporal level consisting in their need for temporally continuous non-interrupted activity as measured by the dynamic INT in our study.

References

Gabard-Durnam, Laurel J., Carol Wilkinson, Kush Kapur, Helen Tager-Flusberg, April R. Levin, and Charles A. Nelson. “Longitudinal EEG Power in the First Postnatal Year Differentiates Autism Outcomes.” *Nature Communications* 10, no. 1 (September 13, 2019): 4188. <https://doi.org/10.1038/s41467-019-12202-9>.

Lopez, K. L., A. D. Monachino, K. M. Vincent, F. C. Peck, and L. J. Gabard-Durnam. “Stability, Change, and Reliable Individual Differences in Electroencephalography Measures: A Lifespan Perspective on Progress and Opportunities.” *NeuroImage* 275 (July 15, 2023): 120116. <https://doi.org/10.1016/j.neuroimage.2023.120116>.

Prochnow, Astrid, Xianzhen Zhou, Foroogh Ghorbani, Paul Wendiggensen, Veit Roessner, Bernhard Hommel, and Christian Beste. “The Temporal Dynamics of How the Brain Structures Natural Scenes.” *Cortex* 171 (February 1, 2024): 26–39. <https://doi.org/10.1016/j.cortex.2023.10.005>.

Stevens, Carl E., and Darya L. Zabelina. “Classifying Creativity: Applying Machine Learning Techniques to Divergent Thinking EEG Data.” *NeuroImage* 219 (October 1, 2020): 116990. <https://doi.org/10.1016/j.neuroimage.2020.116990>.

Zhang, Chenyan, Ann-Kathrin Stock, Moritz Mückschel, Bernhard Hommel, and Christian Beste. “Aperiodic Neural Activity Reflects Metacontrol.” *Cerebral Cortex (New York, N.Y.: 1991)* 33, no. 12 (June 8, 2023): 7941–51. <https://doi.org/10.1093/cercor/bhad089>.

9) Finally, they use the Wilson-Cowan type of firing rate models to simulate neural masses to investigate underlying mechanisms based on recurrent connectivity among neurons potentially; however, how the population firing rates relate to calcium imaging data, which measures fluorescence as a proxy of voltage-gated-ion channels in mouse data, needs to be clarified. This needs to be mentioned clearly in the methods.

Authors: We agree that the link is needed and add the following sentences to the methods section.

Methods

Biophysical Modeling

The firing rates relate to calcium imaging data since the fluorescence of a cell reflects the average activity of the calcium sensors. The activity of the calcium sensors reflects the average calcium concentration over the past few hundred milliseconds. Calcium concentration finally reflects the number of spikes fired by the cell, leading us to firing rates we model via the equations described above (Chen et al., 2013; Dana et al., 2016; Stringer et al., 2019).

References

Chen, Tsai-Wen, Trevor J. Wardill, Yi Sun, Stefan R. Pulver, Sabine L. Renninger, Amy Baohan, Eric R. Schreier, et al. "Ultrasensitive Fluorescent Proteins for Imaging Neuronal Activity." *Nature* 499, no. 7458 (July 18, 2013): 295–300. <https://doi.org/10.1038/nature12354>.

Dana, Hod, Boaz Mohar, Yi Sun, Sujatha Narayan, Andrew Gordus, Jeremy P. Hasseman, Getahun Tsegaye, et al. "Sensitive Red Protein Calcium Indicators for Imaging Neural Activity." *eLife* 5 (March 24, 2016): e12727. <https://doi.org/10.7554/eLife.12727>.

Stringer, Carsen, and Marius Pachitariu. "Computational Processing of Neural Recordings from Calcium Imaging Data." *Current Opinion in Neurobiology, Machine Learning, Big Data, and Neuroscience*, 55 (April 1, 2019): 22–31. <https://doi.org/10.1016/j.conb.2018.11.005>.

10) I think it would be better if, instead of recursively tuning the learning parameter to reach a target excitatory firing rate, they consider using a dynamic equation that up or down-regulates the learning rate depending on the state of the neurons or whether the target rate is achieved. This, I think, theoretically speaking a more accurate formulation than the gradient descent method that was run until the error $\langle x \rangle_{time, rois} - target$ was lower than the tolerance value of 0.02, then run 10 more times.

Authors: We see the excellent point and implemented the determination of external input using an analytical method. We added another equation for the input, determining it dynamically as a control problem. This leads to the following equations (writing down time dependencies explicitly for clarity):

$$\tau \frac{dx_i(t)}{dt} = -x_i(t) + f \left(\sum_j W_{ij} x_j(t) + b + I_i(t) \right) \quad (1)$$

$$\tau_I \frac{dI_i(t)}{dt} = \tilde{I}_i(t) - I_i(t) \quad (2)$$

$\tilde{I}_i(t)$ denotes the desired input for region i to reach the desired firing rate. We denote the targeted firing rate as \bar{x} , which is same for every region in the model. To determine $\tilde{I}_i(t)$, we start with noting $f(x)$ and its inverse $f^{-1}(x)$:

$$f(x) = \frac{1}{1 + e^{-rx}} \quad (3)$$

$$f^{-1}(x) = \frac{\log\left(\frac{x}{1-x}\right)}{r} \quad (4)$$

To calculate $\tilde{I}_i(t)$, we look at the steady state solutions for x_i and replace x_i with \bar{x} since we want $x_i = \bar{x}$ in the steady state. Additionally, we replace $I_i(t)$ with $\tilde{I}_i(t)$ since again, we claim that $\tilde{I}_i(t)$ is the input value that would give us $x_i = \bar{x}$ and based on (2), $I_i(t)$ will approach $\tilde{I}_i(t)$ exponentially with the rate τ_I :

$$\bar{x} = f\left(\sum_{j \neq i} W_{ij}x_j(t) + W_{ii}\bar{x} + b + \tilde{I}_i(t)\right) \quad (5)$$

$$\frac{\log\left(\frac{\bar{x}}{1-\bar{x}}\right)}{r} = \sum_{j \neq i} W_{ij}x_j(t) + W_{ii}\bar{x} + b + \tilde{I}_i(t) \quad (6)$$

$$\tilde{I}_i(t) = \frac{\log\left(\frac{\bar{x}}{1-\bar{x}}\right)}{r} - \sum_{j \neq i} W_{ij}x_j(t) - W_{ii}\bar{x} - b \quad (7)$$

Implementing this numerically led us to the following changes in the methods, results and supplementary material sections:

Methods

Biophysical Modelling

To account for the possible confounding effect of firing rate on timescales, we run control analyses where we set I as a function of time for each value of recurrent connections so that in the resting state, the mean of firing rate across time and regions is 0.1 and for stimulated state, it is equal to 0.6. To achieve this, we modify the equations by including dynamics for I :

$$\tau \frac{dx_i(t)}{dt} = -x_i(t) + f\left(\sum_j W_{ij}x_j(t) + b + I_i(t)\right) \quad (4)$$

$$\tau_I \frac{dI_i(t)}{dt} = \tilde{I}_i(t) - I_i(t) \quad (5)$$

$\tilde{I}_i(t)$ denotes the desired input for region i to reach the desired firing rate. We denote the targeted firing rate as \bar{x} , which is same for every region in the model. To determine $\tilde{I}_i(t)$, we start with noting $f(x)$ and its inverse $f^{-1}(x)$:

$$f(x) = \frac{1}{1 + e^{-rx}} \quad (6)$$

$$f^{-1}(x) = \frac{\log\left(\frac{x}{1-x}\right)}{r} \quad (7)$$

To calculate $\tilde{I}_i(t)$, we look at the steady state solutions for x_i and replace x_i with \bar{x} since we want $x_i = \bar{x}$ in the steady state. Additionally, we replace $I_i(t)$ with $\tilde{I}_i(t)$ since again, we claim that $\tilde{I}_i(t)$ is the input value that would give us $x_i = \bar{x}$ and based on (5), $I_i(t)$ will approach $\tilde{I}_i(t)$ exponentially with the rate τ_I :

$$\bar{x} = f\left(\sum_{j \neq i} W_{ij}x_j(t) + W_{ii}\bar{x} + b + \tilde{I}_i(t)\right) \quad (8)$$

$$\frac{\log\left(\frac{\bar{x}}{1-\bar{x}}\right)}{r} = \sum_{j \neq i} W_{ij}x_j(t) + W_{ii}\bar{x} + b + \tilde{I}_i(t) \quad (9)$$

$$\tilde{I}_i(t) = \frac{\log\left(\frac{\bar{x}}{1-\bar{x}}\right)}{r} - \sum_{j \neq i} W_{ij}x_j(t) - W_{ii}\bar{x} - b \quad (10)$$

Supplementary figures 38 and 39 show the firing rates for all regions and determined values of I . To solve the equations numerically, we set τ_I to 0.05 and reduce the time step to 0.01.

Results

Recurrent Connections mediate rest – stimulus change of INT in the model

Old: It can be argued that this relationship between recurrent connections and τ change is driven by differences in firing rates of the regions rather than by their intraregional recurrent connections. Indeed, when all else is held the same, the average firing rates increase with increasing recurrent connections (supplementary figures 25 and 26). To counter this argument, we fixed the external input for all variations in recurrent connections so that all models show an average (across time and ROIs) firing rate of 0.1 at rest and 0.6 at stimulation using a gradient descent algorithm for values of W_{ii} from 0 to 3 (see methods).

New: It can be argued that this relationship between recurrent connections and τ change is driven by differences in firing rates of the regions rather than by their intraregional recurrent connections. Indeed, when all else is held the same, the average firing rates increase with increasing recurrent connections (supplementary figures 36 and 37). To counter this argument, we fixed the external input for all variations in recurrent connections so that all models show an average (across time and ROIs) firing rate of 0.1 at rest and 0.6 at stimulation values of W_{ii} from 0 to 3 (see methods). We determined the values for external input by solving the model equations for the external input in a steady state where all firing rates are equal to the targeted firing rate.

Supplementary Material

Supplementary Figure 36: Firing rate time series in one simulation for all regions in resting state with optimized value of external stimulation so that average firing rate will be around 0.1. W_{ii} denote recurrent connections. I stands for external input and $\langle x \rangle$ is the average firing rate across time and regions of interest. Each color denotes one region.

Supplementary Figure 37: Firing rate time series in one simulation for all regions in stimulated state with optimized value of external stimulation so that average firing rate will be around 0.6. W_{ii} denote recurrent connections. I stands for external input and $\langle x \rangle$ is the average firing rate across time and regions of interest. Each color denotes one region.

We reran the simulations with these values of external stimulations to get the rest – stimulated state comparison of τ values, obtained using the same way in the main paper. Supplementary figure 38 shows

the values in all simulations. Note the steady increase of the τ difference with increasing recurrent connections.

Supplementary Figure 38: After determining the strength of external stimulation for each value of recurrent connection, we simulate the models again and compare the timescale values in rest and stimulated states. Each dot denotes one region and colors denote simulations.

11) Please explicitly mention that the mathematical model you use has only one variable, which is excitatory. I do not see any inhibitory variable. Typically, when we speak of recurrent connections, it is between E-I coupling in a biophysical sense. Hence, when the authors say we use the large-scale biophysical model to investigate the role of recurrent connections on the flexible rest task changes of the INT, I am not sure which recurrent connections they are talking about. Finally, what differentiates short-range (recurrent) and long-range (typically excitatory) connections is if they use diffusion tensor imaging (DTI) or human connectome data to constrain the anatomical connectivity between relevant ROIs. Many of these things are missing from methods, making it harder to understand and appreciate their model (despite a schematic in Figure 8).

Authors: We thank the reviewer for pointing out these shortcomings. The following changes in the methods will clarify the issues. We used only excitatory regions in the model to make the model as simple as possible; however, we now also replicated the results in a model that incorporates both excitatory and inhibitory connections. The inhibition was incorporated in the model by including one excitatory and one inhibitory population for each region. Following changes in the manuscript and supplementary materials describe the additions we made in the paper.

Methods

Biophysical Modelling

Old: The diagonal elements of W are recurrent connections.

New: The DTI matrix consists only of interareal (long – range) connections, leaving the intra-areal / recurrent connections unspecified. The off – diagonal elements of W are therefore constrained by the DTI matrix. The diagonal elements determine recurrent connections.

Old: We used a firing rate model to model our findings.

New: We used a firing rate model that consists only of excitatory connections to model our findings. The choice of excitatory connections ensures simplicity in the model, avoiding parameters related to inhibition. In addition, we replicated the main result of our paper in a model that also includes inhibitory connections. This is achieved via incorporating one inhibitory and one excitatory population to each region. The model specification and results can be seen in supplementary material section “Model with inhibitory connections”.

Supplementary Material

Model with Inhibitory Connections

In order to replicate our results in a model that also includes inhibitory connections, we modified the equations by incorporating one excitatory and one inhibitory population for each region:

$$\tau_E \frac{dx_{E,i}(t)}{dt} = -x_{E,i}(t) + f \left(\sum_{i \neq j} W_{ij} x_{E,i}(t) + C_{EE} x_{E,i}(t) - C_{EI} x_{I,i}(t) + b + I_i(t) \right)$$
$$\tau_I \frac{dx_{I,i}(t)}{dt} = -x_{I,i}(t) + f(C_{IE} x_{E,i}(t) - C_{II} x_{I,i}(t) + b)$$

Where $x_{E,i}$ and $x_{I,i}$ represent the firing rates of excitatory and inhibitory populations in region i respectively. C_{ij} is the weight matrix of intraareal connections. For numerical simulations, we set $\tau_E=0.1$, $\tau_I=0.05$, $C_{II}=0.1$, $C_{IE}=0.2$, $C_{EI}=0.8$. Remaining parameters are the same as the excitatory-only simulations. By defining C_{EE} as a separate term, we set the diagonal elements of W to 0. C_{EE} corresponds to recurrent excitatory connections and we explore their effect on the dynamics via changing C_{EE} systematically from 0 to 4 in steps of 0.5. Below, we replicate figures 8 and 9 from the main manuscript in this formalism and investigate the ACW changes of

excitatory populations.

Supplementary Figure 41. a. Rest – Stimulation change of timescale for the resting model and stimulated model incorporating inhibitory connections. b. Correlation between rest variability of timescale and rest - task change of timescale.

Supplementary Figure 42. Change of timescale for various values of recurrent connections in the model including inhibitory connections.

Reviewer #2 (Remarks to the Author):

Brief summary

The authors examine how intrinsic timescales (autocorrelation) of neural activity relate to certain types of behavior. They examine calcium activity in behaving mice and EEG activity in humans performing a cognitive task, finding that INT predicts behavioral state and influences future activity in a state-dependent manner. These findings are supported by a RNN model, which shows that INT is effected by auto-connection strength in the weight matrix.

Overall impressions

The finding that INT can predict - and influence - behavior is of significant interest to the field. The authors replicate their result in two different datasets, significantly strengthening their argument. Examining the network mechanisms using a model provides additional predictions for future work for the field and makes the overall work more compelling. Additional work is needed to show the specificity of the effect, and provide additional information and context to understand the model. This work will significantly strengthen the author's arguments.

Specific comments

1. The finding that the autocorrelation, or intrinsic timescale, of regional cortical activity is very interesting. To interpret this finding it is important to characterize the neural activity itself more thoroughly, especially because correlation is influenced by magnitude of the activity. For example, what is the activation rate and variability (e.g. Fano facotr) of neurons in the calcium imaging data? Are there changes in the power spectra across states in the EEG data?

Authors: We thank the reviewer for his positive and encouraging comments. We agree with the reviewer that a more comprehensive evaluation of dynamics is warranted, which we now supply as below. In summary, we replicated the rest – locomotion comparisons in mice for both mean and standard deviation of Calcium Imaging activity for the mice. For human EEG data, we compared the three states' power bands (theta: 1-4 Hz, delta: 4-7 Hz, alpha: 8-12 Hz, beta: 13-30 Hz, gamma: 30-40 Hz, and broadband: 1-40 Hz). We unfortunately could not look for Fano factors in the mice data since the preprocessed data shared by Shamsavarani et al. were averaged across all signals in a region of interest, hence, they lacked any timing detail of individual spikes. Below, we show the additions made in the paper that reflects these analyses.

Methods

Additional Analyses to Further Investigate Neural Dynamics

To characterize the neural activity more thoroughly, we compared a number of features between the behavioral states in mice and human data. In mice data, we compared the mean and standard deviation (SD) of the calcium imaging activity between sustained rest and locomotion states. In human EEG data, we compared the power bands (theta: 1-4 Hz, delta: 4-7 Hz, alpha: 8-12 Hz, beta: 13-30 Hz, gamma: 30-40 Hz, and broadband: 1-40 Hz) across the three states (rest, self, non – self). The mean and SD from mice and the power bands from humans were calculated in the same sliding window fashion as the other measures: we extract

one measure per ROI / channel from each 10 second non – overlapping time window. To extract power bands, we used the periodogram method and calculated the area under the curve of the power spectrum using the trapezoid method.

Results

Changes in Intrinsic Neural Timescales from Rest to Behavior

In addition to intrinsic neural timescales, we also compared mean and standard deviation (SD) of the neural activity to characterize the neural activity more thoroughly. One mean and SD value was extracted from each ROI per 10 second time window of activity, in the same way as the INTs were calculated. We averaged the neural activity across these windows and compared them between sustained rest and locomotion conditions. The results are presented in supplementary figure 11. In short, we observed higher mean of activity in locomotion compared to sustained rest ($z = 25.64$, $p < 0.001$, $r = 0.98$) and higher SD in sustained rest compared to locomotion ($z = 15.53$, $p < 0.001$, $r = -0.59$).

Replication of the Mice Results in a Human EEG Dataset of Self-Evaluation

To further characterize the dynamics of neural activity, we compared EEG power bands between the three states. We extracted one power band per each of the 10 second non – overlapping windows per channel and averaged across these values for comparison. The powers were calculated as the area under the curve of the power spectrum for theta (1-4 Hz), delta (4-7 Hz), alpha (8-12 Hz), beta (13-30 Hz), gamma (30-40 Hz) bands and broadband power (1-40 Hz).

The results are presented in supplementary figures 29 and 30. In short, we observed higher power in resting state compared to self and non – self tasks in all powers, though the effect size rest versus other comparison for gamma band was rather low (theta, rest vs self: $z = 11.53$, $p < 0.001$, $r = -0.26$; rest vs other: $z = 9.11$, $p < 0.001$, $r = -0.2$; delta: rest vs self: $z = 14.06$, $p < 0.001$, $r = -0.31$; rest vs other: $z = 12.45$, $p < 0.001$, $r = -0.28$; alpha: rest vs self: $z = 14.10$, $p < 0.001$, $r = -0.31$, rest vs other: $z = 9.71$, $p < 0.001$, $r = -0.22$; beta: rest vs self: $z = 9.88$, $p < 0.001$, $r = -0.22$; rest vs other: $z = 7.82$, $p < 0.001$, $r = -0.17$; gamma: rest vs self: $z = 5.69$, $p < 0.001$, $r = -0.13$, rest vs other: $z = 3.79$, $p < 0.001$, $r = -0.08$; broadband: rest vs self: $z = 13.43$, $p < 0.001$, $r = -0.30$, rest vs other: $z = 10.25$, $p < 0.001$, $r = -0.23$). Regarding self vs other comparisons, we observed statistically significant higher power in other condition compared to rest in every frequency band, but effect sizes are lower than 0.1 (theta: $z = 2.9$, $p = 0.004$, $r = 0.06$; delta: $z = 2.08$, $p = 0.038$, $r = 0.05$; alpha: $z = 3.54$, $p < 0.001$, $r = 0.08$; beta: $z = 2.14$, $p = 0.032$, $r = 0.05$; gamma: $z = 2.02$, $p = 0.044$, $r = 0.04$; broadband: $z = 3.38$, $p < 0.001$, $r = 0.08$).

Supplementary Material

Comparison of Mean and Standard Deviation of Neural Activity Across Behavioral States in Mice

Supplementary figure 11 shows the rest – locomotion difference of mean and SD of neural activity in mice data.

Supplementary Figure 11. Comparison of mean and SD of neural activity in mice across sustained rest and locomotion conditions.

Comparison of Power Bands in Human EEG Data

In addition to timescales, we also calculated the power bands in human EEG data and compared them across the three conditions. Supplementary figures 29 and 30 show the results and results without outliers, defined as the values which are three median absolute deviations away from the median.

Supplementary Figure 29. Comparison of power bands across three states in human EEG data. Horizontal black lines show median.

Supplementary Figure 30. Comparison of power bands across three states in human EEG data without the outliers which are defined as three median absolute deviations away from the median. Horizontal black lines show median.

References

Shahsavarani, Somayeh, David N. Thibodeaux, Weihao Xu, Sharon H. Kim, Fatema Lodgher, Chinwendu Nwokeabia, Morgan Cambareri, et al. "Cortex-Wide Neural Dynamics Predict Behavioral States and Provide a Neural Basis for Resting-State Dynamic Functional Connectivity." *Cell Reports* 42, no. 6 (June 27, 2023): 112527. <https://doi.org/10.1016/j.celrep.2023.112527>.

2. Because the authors argue for the specific significance of INTs, it's important to demonstrate whether their findings are specific to that metric. Additional comparators are needed. For example, how well does the simple mean or variance of the calcium activity predict behavioral state in the SVM? Similarly, how well does average power predict state in the EEG data? Sub bands?

Authors: We thank the reviewer for this observation. Throughout the paper, we refrain from making an assertion about the specificity of the ACW. Indeed, many indicators of neuronal dynamics can also have explanatory power in terms of behavior. Previous research has shown that neural oscillations (Iemi et al., 2022; Seeber et al., 2016; Liuzzi et al., 2023; Andalman et al., 2019) and variability (Waschke et al., 2021; Cohen and Maunsell, 2009; Garrett et al., 2013; Garrett et al., 2011; Grady et al., 2017; McGinley et al., 2015) also reflect behavioral states. Furthermore, previous research also shows that INT and various aspects of neuronal dynamics are not orthogonal to each other, but are correlated (see for example for intertrial phase coherence: Lechner et al., 2023; for infraslow phase dynamics: Ao et al., 2023; for functional connectivity: Manea et al., 2022; Ito et al., 2020; power – law exponent in power spectrum: Zilio et al., 2021; alpha oscillations: Espinoza et al., 2024).

Our claim is weaker than specificity, we only argue that INT contain some specific information about the behavioral state. In other words, even if the INT and other measures of neuronal dynamics are correlated, these other measures do not make INT redundant. From a statistical perspective, our claim is realized by a two – step analysis: 1) showing potential correlations between INT and variance (for mice data); INT and power bands (for EEG data); 2) showing that the relationship between INT and behavioral state does not change significantly when conditioned on these other dynamic measures (McElreath 2020, chapter 5).

To achieve the second step, we first perform a logistic regression with ACW as predictor and behavioral state as outcome, then do a second regression with ACW and other measures (variance for mice, power bands for human data) and compare the regression coefficients using the statistical test developed by Clogg et al. (1995). Note that we are not testing for the mean in mice data because the mean is subtracted from the time series when ACW is calculated in order to eliminate spurious covariance, therefore it is not a potential confound. We show that the inclusion and thus addition of the other predictors do not change the regression coefficients of ACW significantly. While we cannot say that the regression coefficients stay the same due to the logic of null hypothesis significance testing, we can rule out the claim that the predictive value of ACW is a spurious one in that it solely depends on the measures of the neuronal dynamics we tested. Below are the results in the context of the paper. We also added a paragraph to the discussion section and made our stance clear that the INT is not an explicit measure of the behavioral states themselves, but contain information about and related to them.

Results

Changes in Intrinsic Neural Timescales from Rest to Behavior

It can be argued that the relationship between INT and behavioral states is a spurious one, possibly mediated by some other aspect of neuronal dynamics. Indeed, previous research has

shown that neural variability and oscillations also reflect the behavioral states (Iemi et al., 2022; Seeber et al., 2016; Liuzzi et al., 2023; Andalman et al., 2019; Waschke et al., 2021; Cohen and Maunsell, 2009; Garrett et al., 2013; Garrett et al., 2011; Grady et al., 2017; McGinley et al., 2015). To demonstrate that the relationship between INT and behavioral states is not spurious, we present a two – step analysis: 1) showing potential correlations between INT and variance (for mice data); INT and power bands (for human EEG data); 2) showing that the relationship between INT and behavioral state does not change significantly when conditioned on the standard deviation (SD, normalized variance) of the data (McElreath 2020, chapter 5).

To achieve the second step, we first perform a logistic regression with ACW as predictor and behavioral state as outcome, then do a second regression with ACW and SD and compare the regression coefficients using the statistical test developed by Clogg et al. (1995). The results for the correlation are presented in supplementary figure 12. We observe a negative correlation between SD and ACW (Spearman's $\rho=-0.107$, $p=0.047$). The complete results for the logistic regressions are presented in supplementary tables 1 and 2. Clogg test do not show a significant change in the regression coefficient ($z = 0.0162$, $p=0.987$). While we cannot say that the regression coefficient stays the same due to the logic of null hypothesis significance testing, we can rule out the claim that the predictive value of ACW is a spurious one in that it solely depends on the measures of the neuronal dynamics we tested, that is, that the relationship of ACW and behavioral state is spurious.

Replication of the Mice Results in a Human EEG Dataset of Self-Evaluation

As stated above for mice, an argument can be made against the relationship between INT and behavioral states by asserting that the relationship is spurious, confounded by neural oscillations since neural oscillations were also found to reflect behavioral states (Iemi et al., 2022; Seeber et al., 2016; Liuzzi et al., 2023; Andalman et al., 2019). To counter the argument of spurious relationship, we repeat the analyses for the mice: we first investigate the correlations between ACW and neural oscillations (theta, delta, alpha, beta, gamma and broadband) and then condition the relationship between ACW and these oscillations by adding them into the logistic regression between ACW and behavioral state. If the regression coefficient is significantly different, it is possible that the relationship is spurious.

The results for correlation analyses are shown in supplementary figure 31. Briefly, we observe a weak and positive correlation between INT and theta oscillatory power ($\rho=0.069$, $p<0.001$) and negative correlations between INT and other power bands (delta: $\rho=-0.181$, $p<0.001$; alpha: $\rho=-0.494$, $p<0.001$; beta: $\rho=-0.415$, $p<0.001$; gamma: $\rho=-0.262$, $p<0.001$; broadband: $\rho=-0.359$, $p<0.001$). The results for logistic regressions are presented in supplementary tables 3 and 4. We did not observe a significant change in the regression coefficient when it was conditioned on oscillatory powers ($z=0.724$, $p=0.469$). These results show that the relationship of INT and behavior does not solely depend on these additional measures of oscillatory dynamics, that is, that the INT-behavior relationship is not spurious.

Discussion

Behavioral states reflect INT dynamics

We should note that our claim about the relationship between INT and behavioral states are not specific and exclusive to INT. Previous research have shown that neural oscillations (Iemi et al., 2022; Seeber et al., 2016; Liuzzi et al., 2023; Andalman et al., 2019) and variability (Waschke et

al., 2021; Cohen and Maunsell, 2009; Garrett et al., 2013; Garrett et al., 2011; Grady et al., 2017; McGinley et al., 2015) also reflect behavioral states. In our supplementary analyses, we have shown that the relationship between INT and behavioral states is not spurious and confounded by variance (in mice) or oscillatory power (in human data). Our analysis reflects our claim that INT incorporates some specific information about behavioral states which can't be completely explained by or reduced to the neural activity's variance or oscillatory power.

Supplementary Material

Additional Control Analyses for the Specificity of INT in Mice Data

To counter the argument that variance of the data might be a confounding factor for the behavioral specificity of INT, we performed a two – step analysis: first we showed the correlation between variance and INT, second, we used the variance as a confounder in a regression analysis. Supplementary figure 12 shows the correlation whereas supplementary tables 1 and 2 shows the results for logistic regression.

Supplementary Figure 12. Correlation between ACW and SD in mice data

Supplementary table 1. Logistic regression between behavioral state and ACW in mice calcium imaging data

Name	Estimate	SE	tStat	DoF	p value	5% CI	95% CI
Intercept	1.935	0.200	9.665	343	<0.001	1.541	2.329
ACW	-2.441	0.550	-4.432	343	<0.001	-3.525	-1.358

Supplementary table 2. Logistic regression between behavioral state and ACW + SD in mice calcium imaging data

Name	Estimate	SE	tStat	DoF	p value	5% CI	95% CI
Intercept	1.953	0.212	9.193	342	<0.001	1.535	2.372
ACW	-2.454	0.553	-4.437	342	<0.001	-3.541	-1.366
SD	-0.791	2.919	-0.271	342	0.786	-6.534	4.951

Additional Control Analyses for the Specificity of INT in EEG Data

To counter the argument that the power bands in EEG data might be a confounding factor for the behavioral specificity of INT, we performed a two – step analysis: first we showed the correlation between power bands and INT, second, we used the power bands as a confounder in a regression analysis. Supplementary figure 31 shows the correlation whereas supplementary tables 3 and 4 shows the results for logistic regression.

Supplementary Figure 31. Correlation between oscillatory power bands and INT.

Supplementary table 3. Logistic regression between behavioral state and INT in human EEG data

Name	Estimate	SE	tStat	DoF	p value	5% CI	95% CI
Intercept	0.664	0.007	86.03	189500	0	0.649	0.679
INT	0.637	0.152	4.171	189500	<0.001	0.338	0.936

Supplementary table 4. Logistic regression between behavioral state and INT + power bands in human EEG data

Name	Estimate	SE	tStat	DoF	p value	5% CI	95% CI
Intercept	0.724	0.014	50.556	189500	0	0.696	0.752

INT	0.471	0.170	2.760	189500	0.005	0.136	0.806
Delta	-0.012	0.002	-5.963	189500	<0.001	-0.017	-0.008
Theta	0.013	0.003	4.139	189500	<0.001	0.006	0.019
Alpha	-0.007	0.001	-4.063	189500	<0.001	-0.010	-0.003
Beta	-0.003	0.003	-1.296	189500	0.195	-0.009	0.002
Gamma	0.022	0.018	1.247	189500	0.212	-0.012	0.058
Broadband	0.001	0.001	0.083	189500	0.933	-0.003	0.003

References

- Andalman, Aaron S., Vanessa M. Burns, Matthew Lovett-Barron, Michael Broxton, Ben Poole, Samuel J. Yang, Logan Grosenick, et al. "Neuronal Dynamics Regulating Brain and Behavioral State Transitions." *Cell* 177, no. 4 (May 2, 2019): 970-985.e20. <https://doi.org/10.1016/j.cell.2019.02.037>.
- Cohen, Marlene R., and John H. R. Maunsell. "Attention Improves Performance Primarily by Reducing Interneuronal Correlations." *Nature Neuroscience* 12, no. 12 (December 2009): 1594–1600. <https://doi.org/10.1038/nn.2439>.
- Garrett, Douglas D., Natasa Kovacevic, Anthony R. McIntosh, and Cheryl L. Grady. "The Importance of Being Variable." *Journal of Neuroscience* 31, no. 12 (March 23, 2011): 4496–4503. <https://doi.org/10.1523/JNEUROSCI.5641-10.2011>.
- Garrett, Douglas D., Gregory R. Samanez-Larkin, Stuart W. S. MacDonald, Ulman Lindenberger, Anthony R. McIntosh, and Cheryl L. Grady. "Moment-to-Moment Brain Signal Variability: A next Frontier in Human Brain Mapping?" *Neuroscience & Biobehavioral Reviews* 37, no. 4 (May 1, 2013): 610–24. <https://doi.org/10.1016/j.neubiorev.2013.02.015>.
- Grady, Cheryl L., and Douglas D. Garrett. "Brain Signal Variability Is Modulated as a Function of Internal and External Demand in Younger and Older Adults." *NeuroImage* 169 (April 1, 2018): 510–23. <https://doi.org/10.1016/j.neuroimage.2017.12.031>.
- Iemi, Luca, Laura Gwilliams, Jason Samaha, Ryszard Auksztulewicz, Yael M Cycowicz, Jean-Remi King, Vadim V Nikulin, et al. "Ongoing Neural Oscillations Influence Behavior and Sensory Representations by Suppressing Neuronal Excitability." *NeuroImage* 247 (February 15, 2022): 118746. <https://doi.org/10.1016/j.neuroimage.2021.118746>.
- Liuzzi, Lucrezia, Daniel S. Pine, Nathan A. Fox, and Bruno B. Averbeck. "Changes in Behavior and Neural Dynamics across Adolescent Development." *Journal of Neuroscience* 43, no. 50 (December 13, 2023): 8723–32. <https://doi.org/10.1523/JNEUROSCI.0462-23.2023>.
- McGinley, Matthew J., Martin Vinck, Jacob Reimer, Renata Batista-Brito, Edward Zagher, Cathryn R. Cadwell, Andreas S. Tolias, Jessica A. Cardin, and David A. McCormick. "Waking State: Rapid Variations Modulate Neural and Behavioral Responses." *Neuron* 87, no. 6 (September 23, 2015): 1143–61. <https://doi.org/10.1016/j.neuron.2015.09.012>.
- Seeber, Martin, Reinhold Scherer, and Gernot R. Müller-Putz. "EEG Oscillations Are Modulated in Different Behavior-Related Networks during Rhythmic Finger Movements." *Journal of Neuroscience* 36, no. 46 (November 16, 2016): 11671–81. <https://doi.org/10.1523/JNEUROSCI.1739-16.2016>.

Waschke, Leonhard, Niels A. Kloosterman, Jonas Obleser, and Douglas D. Garrett. "Behavior Needs Neural Variability." *Neuron* 109, no. 5 (March 3, 2021): 751–66. <https://doi.org/10.1016/j.neuron.2021.01.023>.

Clogg, Clifford, Αδαμάντιος Χαρίτου, Adamantios Haritou, and Eva Petkova. "Statistical Methods for Comparing Regression Coefficients Between Models." *American Journal of Sociology* 100 (March 1, 1995). <https://doi.org/10.1086/230638>.

McElreath, Richard. *Statistical Rethinking: A Bayesian Course with Examples in R and STAN*. 2 edition. Boca Raton London New York: Chapman and Hall/CRC, 2020.

Ao, Yujia, Yasir Catal, Stephan Lechner, Jingyu Hua, and Georg Northoff. "Intrinsic Neural Timescales Relate to the Dynamics of Infralow Neural Waves." *NeuroImage* 285 (January 2024): 120482. <https://doi.org/10.1016/j.neuroimage.2023.120482>.

Espinosa, Ezequiel Pablo, Di Zang, Andrea Buccellato, Zengxin Qi, Xuehai Wu, Samira Abbasi, Yasir Catal, Stephan Lechner, Federico Zilio, and Georg Northoff. "Spectral Peak Analysis and Intrinsic Neural Timescales as Markers for the State of Consciousness." *NeuroImage: Clinical*, October 30, 2024, 103698. <https://doi.org/10.1016/j.nicl.2024.103698>.

Lechner, Stephan, and Georg Northoff. "Prolonged Intrinsic Neural Timescales Dissociate from Phase Coherence in Schizophrenia." *Brain Sciences* 13, no. 4 (April 21, 2023): 695. <https://doi.org/10.3390/brainsci13040695>.

Manea, Ana MG, Anna Zilverstand, Kamil Ugurbil, Sarah R. Heilbronner, and Jan Zimmermann. "Intrinsic Timescales as an Organizational Principle of Neural Processing across the Whole Rhesus Macaque Brain." *eLife* 11 (March 2, 2022): e75540. <https://doi.org/10.7554/eLife.75540>.

Zilio, Federico, Javier Gomez-Pilar, Shumei Cao, Jun Zhang, Di Zang, Zengxin Qi, Jiaying Tan, et al. "Are Intrinsic Neural Timescales Related to Sensory Processing? Evidence from Abnormal Behavioral States." *NeuroImage* 226 (February 1, 2021): 117579. <https://doi.org/10.1016/j.neuroimage.2020.117579>.

Ito, Takuya, Luke J. Hearne, and Michael W. Cole. "A Cortical Hierarchy of Localized and Distributed Processes Revealed via Dissociation of Task Activations, Connectivity Changes, and Intrinsic Timescales." *NeuroImage* 221 (November 1, 2020): 117141. <https://doi.org/10.1016/j.neuroimage.2020.117141>.

3. There is a significant amount of research on the chaotic nature of RNNs. As the concept of an intrinsic timescale boils down to the autocorrelation of the data and this the stability of the network, the authors should cite some of it. For example Dean Buonomano, Larry Abbott, and Kanaka Rajan have all investigated this issue.

Authors: We thank the reviewer for directing our attention to the notion of chaos in recurrent neural networks. We added the following paragraph on the limitations to reflect the studies on chaos in recurrent neural networks.

Limitations

We should note that while our research did not incorporate the notion of chaos explicitly, there is ample evidence of chaos in the dynamics of recurrent neural networks (Sompolinsky et al., 1988). Research has shown that chaos, and more specifically edge of chaos increases the ability of recurrent neural networks to learn to generate complex periodic patterns of activity from input (Sussillo and Abbott, 2009) and represent timing (Laje and Buonomano, 2013; Hardy et al., 2018). Furthermore, it was shown that external inputs can suppress these chaotic dynamics (Rajan, Abbott and Sompolinsky, 2010). Our future research direction will involve harnessing the chaotic dynamics of recurrent neural networks for the spontaneous generation of behavioral states and how the chaoticity of neural networks impact intrinsic neural timescales.

References

- Sompolinsky, H., A. Crisanti, and H. J. Sommers. "Chaos in Random Neural Networks." *Physical Review Letters* 61, no. 3 (July 18, 1988): 259–62. <https://doi.org/10.1103/PhysRevLett.61.259>.
- Sussillo, David, and L. F. Abbott. "Generating Coherent Patterns of Activity from Chaotic Neural Networks." *Neuron* 63, no. 4 (August 27, 2009): 544. <https://doi.org/10.1016/j.neuron.2009.07.018>.
- Laje, Rodrigo, and Dean V. Buonomano. "Robust Timing and Motor Patterns by Taming Chaos in Recurrent Neural Networks." *Nature Neuroscience* 16, no. 7 (July 2013): 925–33. <https://doi.org/10.1038/nn.3405>.
- Hardy, Nicholas F., Vishwa Goudar, Juan L. Romero-Sosa, and Dean V. Buonomano. "A Model of Temporal Scaling Correctly Predicts That Motor Timing Improves with Speed." *Nature Communications* 9, no. 1 (November 9, 2018): 4732. <https://doi.org/10.1038/s41467-018-07161-6>.
- Rajan, Kanaka, L. F. Abbott, and Haim Sompolinsky. "Stimulus-Dependent Suppression of Chaos in Recurrent Neural Networks." *Physical Review E* 82, no. 1 (July 7, 2010): 011903. <https://doi.org/10.1103/PhysRevE.82.011903>.

4. The stability (or chaotic nature) of the RNN would impact the autocorrelation of the simulated data. The primary motivation of the simulation is to examine the network properties (e.g. connection strength) that may drive the change in autocorrelation. Without some additional characterization of the network structure and dynamics it is difficult to interpret the results of the simulation. For example, the authors could examine the eigenvalues of the network's weight matrix or measure the Lyapunov exponent of the network trajectories.

Authors: The chaotic nature of RNNs is indeed an important potential variable that might influence the INTs. In line with comments 3 and 5, the reviewer makes an excellent recommendation. Following their advice, we analyzed the Lyapunov exponents of the trajectories generated from our RNN simulations. We found that our model does not exhibit chaotic behavior, indicated by negative Lyapunov exponents for all values of recurrent connectivity, in both rest and stimulated states. Below, we present the methodology and results of this analysis, along with Lyapunov spectra from the RNNs.

Methods

Biophysical Modeling

It is important to note that RNNs can exhibit chaotic behavior which can potentially influence INTs. To eliminate the potential confounding nature of chaos, we investigated the Lyapunov spectra (Lyapunov, 1992) of our model. To calculate Lyapunov exponents, we used the ChaosTools.jl package (Datseris and Parlitz, 2022) which is part of the larger ecosystem of DynamicalSystems.jl. The software we used uses H2 algorithm introduced in Geist et al., 1990. Briefly, this method uses QR decomposition on D – dimensional deviation vectors of tangent dynamical system, where D is the dimensionality of the model (in our case, 360 corresponding to 360 regions of interest). The QR decomposition at each step yields the local growth rate of each dimension. This procedure is done through N steps and the growth rates are averaged across these N steps, yielding the Lyapunov exponent for each dimension. In our case, N was chosen to be 5000, with $dt = 0.01$, yielding 50 seconds of simulation. The Lyapunov spectra for each value of recurrent connections and rest versus stimulated states are given in supplementary figure 43.

Results

Recurrent Connections mediate rest – stimulus change of INT in the model

There is ample research that shows chaos in recurrent neural networks, and it can exhibit chaoticity which is a potential influencer on the timescales that we estimate (Sompolinsky et al., 1988; Sussillo and Abbott, 2009; Laje and Buonomano, 2013; Hardy et al., 2018; Rajan, Abbott and Sompolinsky, 2010). To disentangle the potential relationship between INT and chaos, we calculated the Lyapunov spectra of our model in each configuration of recurrent connectivity and input strength. The Lyapunov spectra are shown in supplementary figure 43. None of the Lyapunov exponents in any configuration were positive, indicating that our model does not show chaotic behavior making it rather unlikely that chaos is a potential confounder in our analysis.

Supplementary Material

Lyapunov Spectra of the Model

In order to further characterize the model and assess potential chaoticity of the dynamics, we calculated the Lyapunov spectra for each value of the recurrent connection in both rest and stimulated states. Supplementary figure 43 shows the Lyapunov Spectra.

Supplementary Figure 43. Lyapunov spectra of the model in rest and stimulated states for each value of recurrent connections

Additional References

Datseris, George, and Ulrich Parlitz. *Nonlinear Dynamics: A Concise Introduction Interlaced with Code. Undergraduate Lecture Notes in Physics*. Cham: Springer International Publishing, 2022. <https://doi.org/10.1007/978-3-030-91032-7>.

Geist, Karlheinz, Ulrich Parlitz, and Werner Lauterborn. "Comparison of Different Methods for Computing Lyapunov Exponents." *Progress of Theoretical Physics* 83, no. 5 (May 1, 1990): 875–93. <https://doi.org/10.1143/PTP.83.875>.

Lyapunov, A. M. "The General Problem of the Stability of Motion." *International Journal of Control*, March 1, 1992. <https://doi.org/10.1080/00207179208934253>.

5. The authors use the model to support one their primary arguments: that the timescales of neural activity relate to – and predict – states of motor behavior. However, it is unclear whether the RNN developed in the paper could itself drive motor behavior. This would additionally impact the network’s sensitivity to input strength. Developing a model that can simulate motor activity would demonstrate that it approximates cortical activity, improve its validity, and significantly strengthen the argument of the paper.

For examples see:

Sussillo, D., and Abbott, L.F. (2009). Generating coherent patterns of activity from chaotic neural networks. *Neuron* 63, 544–557. <https://doi.org/10.1016/j.neuron.2009.07.018>.

Laje, R., and Buonomano, D.V. (2013). Robust timing and motor patterns by taming chaos in recurrent neural networks. *Nat. Neurosci.* 16, 925–933. <https://doi.org/10.1038/nn.3405>.

Rajan, K., Abbott, L.F., and Sompolinsky, H. (2010). Stimulus-dependent suppression of chaos

in recurrent neural networks. Phys Rev E Stat Nonlin Soft Matter Phys 82, 011903. <https://doi.org/10.1103/PhysRevE.82.011903>.

Hardy, N.F., Goudar, V., Romero-Sosa, J.L., and Buonomano, D.V. (2018). A model of temporal scaling correctly predicts that motor timing improves with speed. Nature Communications 9, 4732. <https://doi.org/10.1038/s41467-018-07161-6>.

Authors: We thank the reviewer for this excellent suggestion and references they provide. Unfortunately, developing a model that spontaneously generates states of motor behavior is out of scope for our current paper. The trilateral relationship between intrinsic timescales, chaos and behavioral states is a very exciting, timely and bountiful research avenue which would be too comprehensive to fit in our current paper. We note that this will provide a future research avenue for us. To reflect this, we added the following in the limitations section of the paper (which was also written in the answer to the point 3):

Limitations

Our future research direction will involve harnessing the chaotic dynamics of recurrent neural networks (Sompolinsky et al., 1988; Sussillo and Abbott, 2009; Laje and Buonomano, 2013; Hardy et al., 2018; Rajan, Abbott and Sompolinsky, 2010) for spontaneous generation of behavioral states and how the chaoticity of neural networks impact intrinsic neural timescales.